# Convergent and divergent brain structural and functional abnormalities associated with developmental dyslexia

Xiaohui Yan[1], Ke Jiang[1], Hui Li[2], Ziyi Wang[3], Kyle Perkins[4], Fan Cao[1]*

[1]Department of Psychology, Sun Yat-Sen University, Guangzhou, China; [2]Department of Preschool Education, Anyang Preschool Education College, Anyang, China; [3]School of Foreign Language, Jining University, Jining, China; [4]Florida International University (Retired Professor), Miami, United States

**Abstract** Brain abnormalities in the reading network have been repeatedly reported in individuals with developmental dyslexia (DD); however, it is still not totally understood where the structural and functional abnormalities are consistent/inconsistent across languages. In the current multi-modal meta-analysis, we found convergent structural and functional alterations in the left superior temporal gyrus across languages, suggesting a neural signature of DD. We found greater reduction in grey matter volume and brain activation in the left inferior frontal gyrus in morpho-syllabic languages (e.g. Chinese) than in alphabetic languages, and greater reduction in brain activation in the left middle temporal gyrus and fusiform gyrus in alphabetic languages than in morpho-syllabic languages. These language differences are explained as consequences of being DD while learning a specific language. In addition, we also found brain regions that showed increased grey matter volume and brain activation, presumably suggesting compensations and brain regions that showed inconsistent alterations in brain structure and function. Our study provides important insights about the etiology of DD from a cross-linguistic perspective with considerations of consistency/inconsistency between structural and functional alterations.

*For correspondence: caofan3@mail.sysu.edu.cn

**Competing interest:** The authors declare that no competing interests exist.

## Introduction

Individuals with developmental dyslexia (DD) encounter difficulty in learning to read even with normal intelligence and adequate educational guidance (*Peterson and Pennington, 2012*). DD affects a large number of individuals across writing systems, and the prevalence is about 5–10% in alphabetic writing systems (e.g. English and German) (*Döhla and Heim, 2015*; *Katusic et al., 2001*; *Shaywitz, 1996*) and about 4–7% in morpho-syllabic writing systems (e.g. Chinese and Japanese Kanji) (*Sun et al., 2013*; *Uno et al., 2009*; *Zhao et al., 2016*). Multiple deficits have been identified to be associated with DD (*Ring and Black, 2018*), among which phonological deficit has been well-documented across languages (*Gu and Bi, 2020*; *Snowling and Melby-Lervag, 2016*). Individuals with DD show deficient phonological ability including phonological representation, manipulation, and retrieval even when compared to reading-level controls (*Melby-Lervag et al., 2012*; *Parrila et al., 2020*). However, the common phonological deficit may manifest differently in reading behavior depending on the specific requirements of the writing system. For example, phonological deficit in English is associated with lower accuracy in phonological decoding (*Landerl et al., 1997*; *Ziegler et al., 2003*), and it is associated with slower reading speed in transparent orthographies with relatively intact accuracy in phonological decoding (*Wimmer and Schurz, 2010*). In Chinese, phonological deficit is associated with a higher rate of semantic errors during character reading (*Shu et al., 2005*), because children with DD over-rely on the semantic cue in the character during reading due to the inability to use

the phonological cue. According to research, 80 % of Chinese characters have a semantic radical and a phonetic radical providing semantic cues and phonological cues of the character, respectively (*Honorof and Feldman, 2006*).

At the neurological level, the reading network in the left hemisphere has often been found to show alterations in individuals with DD (*Pugh et al., 2000*; *Richlan, 2012*; *Richlan, 2014*), including the temporoparietal cortex (TP), the occipitotemporal cortex (OT), and the inferior frontal cortex. The left TP area is further subdivided into the posterior superior temporal gyrus (STG), which is involved in fine phonological analysis (*Petersen and Fiez, 1993*; *Richlan, 2012*) and the inferior parietal lobule (IPL) which is associated with general attention control (*Richlan, 2014*). This TP region tends to show reduced brain activation in individuals with DD in alphabetic languages as demonstrated in a cross-linguistic study of English, Italian, and French (*Paulesu et al., 2001*) and several meta-analysis studies in alphabetic languages (*Maisog et al., 2008*; *Martin et al., 2016*; *Paulesu et al., 2014*; *Richlan et al., 2009*; *Richlan et al., 2011*). The left OT area, including the middle occipital gyrus (MOG), inferior temporal gyrus (ITG) and fusiform gyrus, has been consistently found to show reduced activation in individuals with DD across morpho-syllabic and alphabetic languages (*Bolger et al., 2008*; *Cao et al., 2020*; *Centanni et al., 2019*; *Chyl et al., 2019*; *Paz-Alonso et al., 2018*). This region is associated with visuo-orthographic processing during reading (*Glezer et al., 2016*; *Glezer et al., 2019*). The left inferior frontal cortex is further subdivided into the inferior frontal gyrus (IFG) and the precentral gyrus (*Richlan, 2014*). The precentral gyrus may relate to compensatory articulatory processes in dyslexia (*Hancock et al., 2017*), whereas the IFG has been known to be involved in phonological and semantic retrieval, lexical selection and integration (*Booth et al., 2007a*; *Booth et al., 2007b*; *Costa-freda et al., 2006*; *Szatkowska et al., 2000*). However, the nature of dysfunction in the left IFG in individuals with DD remains controversial. Although reduced activation in the left IFG was confirmed by many fMRI studies and meta-analysis studies (*Booth et al., 2007a*; *Cao et al., 2006*; *Richlan et al., 2010*; *Wimmer et al., 2010*), increased activation in the left IFG was also reported in many fMRI studies (*Grunling et al., 2004*; *Kronbichler et al., 2006*; *Waldie et al., 2013*; *Wimmer et al., 2010*). The inconsistent results may be related to task and task difficulty (*Waldie et al., 2013*; *Wimmer et al., 2010*), orthographic transparency (*Martin et al., 2016*), and age of participants (*Chyl et al., 2019*).

There is a sparsity in research investigating whether the deficits associated with DD are language-universal. *Paulesu et al., 2001* found that readers with DD in English, Italian, and French showed similar brain abnormality during an explicit word reading task and an implicit reading task. *Hu et al., 2010* found that Chinese and English children with DD showed language-universal deficits. *Feng et al., 2020* found that children with DD in both Chinese and French showed common reduction of brain activation in the left fusiform gyrus and STG. In summary, meta-analytic studies should make a greater contribution in such a topic by gathering studies from different languages and comparing them.

Even though language-universal deficits in the brain have been suggested in several studies (*Feng et al., 2020*; *Hu et al., 2010*; *Paulesu et al., 2001*), language specificity has been demonstrated as well (*Martin et al., 2016*; *Siok et al., 2004*). In a meta-analysis study (*Martin et al., 2016*), researchers directly compared brain deficits associated with DD between transparent and opaque orthographies and found that functional abnormalities in the brain vary with orthographic depth in alphabetic languages. Specifically, consistent reduction of brain activation was found in a left OT area regardless of orthographic depth, whereas greater reduction was found in the left fusiform gyrus, left TP and left IFG pars orbitalis in transparent orthographies than in opaque orthographies, and greater reduction in the bilateral intraparietal sulcus, left precuneus and left IFG pars triangularis was found in opaque orthographies than in transparent orthographies. In a recent study on Chinese-English bilingual children with DD, researchers also found both language-universal and language-specific deficits for Chinese and English (*Cao et al., 2020*). These findings suggest that there are both language-universal and language-specific deficits across languages. The language-universal deficits might be related to the causal risk of DD while the language-specific deficits tend to be interpreted as a result of interaction between DD and the specific language system that one studies.

Alphabetic and morpho-syllabic languages make a contrastive cross-linguistic comparison. As a representative morpho-syllabic language, Chinese character represents a morpheme and a syllable rather than a phoneme, even though a small percent of Chinese characters are logographic. Research on DD in Chinese has revealed different patterns of brain abnormalities from alphabetic languages.

Significant alteration in the left middle frontal gyrus (MFG) or dorsal IFG has been consistently reported in different studies (*Cao et al., 2017*; *Cao et al., 2020*; *Liu et al., 2012*; *Liu et al., 2013a*; *Siok et al., 2004*; *Siok et al., 2008*), while the alteration in the left TP areas has been reported in only a few studies (*Cao et al., 2017*; *Cao et al., 2018*; *Hu et al., 2010*). This might be because the left dorsal IFG plays an essential role in Chinese. It has been found that the left dorsal IFG is more involved in Chinese reading than in English reading, while the left TP is more involved in English reading than in Chinese reading in typical readers (*Bolger et al., 2005*; *Tan et al., 2005*). Therefore, the greater deficit in the left dorsal IFG suggests a Chinese-specific deficit.

In addition to functional studies, there have also been a large number of studies with a focus on structural alterations associated with DD. Even though brain structural alterations may cause DD, it is equally possible that altered brain structure is a result of being DD, since learning experience shapes brain development. However, only one of these studies has taken language difference into account (*Silani et al., 2005*). Previous meta-analysis studies on alphabetic languages have found grey matter reduction in the left TP area (*Linkersdörfer et al., 2012*; *McGrath and Stoodley, 2019*; *Richlan et al., 2013*) as well as the left OT area (*Linkersdörfer et al., 2012*). These three meta-analytic studies echo findings from functional studies by showing abnormal brain structures within the classic reading network in alphabetic languages. In consistent, studies on morpho-syllabic languages have also found grey matter reduction within the classic reading network (*Liu et al., 2013b*; *Siok et al., 2008*; *Wang et al., 2019*; *Xia et al., 2016*). However, there are also studies that found abnormal brain structures outside the classic reading network, for example, in putamen, cerebellum, thalamus, and caudate etc. (*Adrian-Ventura et al., 2020*; *Brambati et al., 2004*; *Brown et al., 2001*; *Jagger-Rickels et al., 2018*; *Jednorog et al., 2015*; *Wang et al., 2019*), suggesting that these regions are also affected in this condition. Taken together, both classic reading regions and other regions have been found to show structural alterations in DD, and the previous inconsistent findings in brain structure might be due to the lack of differentiation in participants' language. It is important to differentiate language-universal structural alterations as a core deficit which might be related to the cause of DD and language-specific structural alterations as a consequence of being DD in a specific language. Learning a specific language with DD may affect brain development in regions that are specifically important for that language and related functions (*Mechelli et al., 2004*).

DD is associated with altered brain structure and function, but very few studies have investigated whether brain structural and functional alterations are consistent or inconsistent. In a study by *Siok et al., 2008*, researchers examined both structural and functional alterations in Chinese children with DD, and found reduced GMV and brain activation in the left MFG, which underscores the association between the left MFG and DD in Chinese. Another study located a key region in the left IPL, which showed reduced GMV and activation in English-speaking readers with DD (*Hoeft et al., 2007*). A recent study, from a developmental perspective, found a dissociation between brain structural development and brain functional development in some brain regions (e.g. left fusiform gyrus) in the context of reading development (*Siok et al., 2020*), suggesting that learning experience may significantly shape brain function independent of brain structure. However, more research is sorely needed to examine whether there are brain regions that show increased GMV but decreased brain function or vice versa, and to understand the neurocognitive implications of such patterns. Simultaneously considering structural and functional abnormalities with a focus on cross-linguistic comparison would provide a comprehensive perspective to understand the neural mechanisms of DD.

In this meta-analysis study, we aimed to explore how structural and functional impairment of DD converge or diverge and whether this pattern is similar or different across writing systems. We expected to find brain regions that show decreased brain structure and function, indicating insufficient neuronal resources for certain cognitive computations. For regions that show increased brain structure and function, we believe they develop to an unusually high degree for compensation. For brain regions with increased structure but decreased function or decreased structure and increased function, it may be due to brain structures receiving inhibitory input from other regions. We also expected to find language-universal as well as language-specific neurological abnormalities. For language-universal deficits, we tend to believe that they are related to the cause of DD, while the language-specific deficits tend to be consequences of DD in different languages.

**Table 1.** Functional deficits in individuals with DD in alphabetic languages (JK represents the results of jack-knife sensitivity analysis).

| Regions | MNI coordinate | SDM-Z | p | Voxels | Cluster breakdown (Voxels) | JK |
|---|---|---|---|---|---|---|
| *Hypoactivation in DD* | | | | | | |
| Left supramarginal gyrus | −56,−46,30 | 4.919 | 0.0000 | 10,742 | Left IPL, BA 40 (986)<br>Left MTG, BA 21 (719)<br>Left MTG, BA 37 (634)<br>Left ITG, BA 37 (601)<br>Left fusiform gyrus, BA 37 (479)<br>Left ITG, BA 20 (414)<br>Left STG, BA 48 (377)<br>Left supramarginal gyrus, BA 48 (329)<br>Left angular gyrus, BA 39 (307)<br>Left MTG, BA 22 (300)<br>Left cerebellum, lobule VI, BA 37 (285)<br>Left STG, BA 42 (281)<br>Left rolandic operculum, BA 48 (261)<br>Left arcuate network (255)<br>Left cerebellum, crus I, BA 37 (210)<br>Left STG, BA 22 (201)<br>Left supramarginal gyrus, BA 40 (198)<br>Left superior longitudinal fasciculus III (182)<br>Left IPL, BA 2 (167)<br>Left inferior occipital gyrus, BA 19 (156) | 79/79 |
| Right MOG | 42,−86,6 | 2.244 | 0.0002 | 361 | Right MOG, BA19 (208) | 76/79 |
| Right STG | 60,−16,4 | 1.936 | 0.0011 | 358 | | 73/79 |
| *Hyperactivation in DD* | | | | | | |
| Right cerebellum | 26,−60,−28 | −1.526 | 0.0000 | 1,559 | Right cerebellum, lobule VI, BA 37 (352)<br>Right cerebellum, lobule VI, BA 19 (233)<br>Middle cerebellar peduncles (212) | 79/79 |
| Left caudate nucleus | −16,12,6 | −1.459 | 0.0000 | 611 | Left anterior thalamic projections (364) | 79/79 |
| Right caudate nucleus | 10,2,14 | −1.317 | 0.0001 | 520 | Right anterior thalamic projections (185)<br>Right caudate nucleus (184) | 79/79 |

# Results

## Description of the included studies

For the functional studies, a total of 2728 participants (controls:1370, DD:1358) were included, and the mean age was 16.56 years for controls and 16.26 years for participants with DD. Specifically, there were 79 functional experiments in alphabetic languages, including 31 experiments on adults (N = 434, mean age = 26.12 for controls, N = 411, mean age = for 25.86 for DD), 36 experiments on children (N = 553, mean age = 10.59 for controls, N = 586, mean age = for 10.54 for DD), 7 experiments on adolescents (N = 131, mean age = 14.44 for controls, N = 108, mean age = for 14.30 for DD), and 5 studies of mixed ages. There were 12 functional experiments in morpho-syllabic languages (N = 164, mean age = 11.48 for controls, N = 162, mean age = for 11.45 for DD), including 11 experiments on children and 1 experiment on adolescents.

For the structural studies, there were 21 experiments in alphabetic languages, including 10 experiments on adults (N = 209, mean age = 26.68 for controls, N = 193, mean age = for 27.15 for DD), 8 experiments on children (N = 245, mean age = 10.17 for controls, N = 266, mean age = for 10.28 for DD), 1 experiment on adolescents and 2 studies of mixed ages. There were six structural experiments on children in morpho-syllabic languages (N = 89, mean age = 11.82 for controls, N = 94, mean age = for 11.74 for DD).

## Meta-analysis results

### Functional deficits in alphabetic languages and morpho-syllabic languages

In the meta-analysis of functional studies in alphabetic languages, hypoactivation in DD was found in a large cluster peaked at the left supramarginal gyrus which extended to the inferior frontal cortex, occipitotemporal cortex and cerebellum, a cluster peaked at the right MOG and a cluster peaked

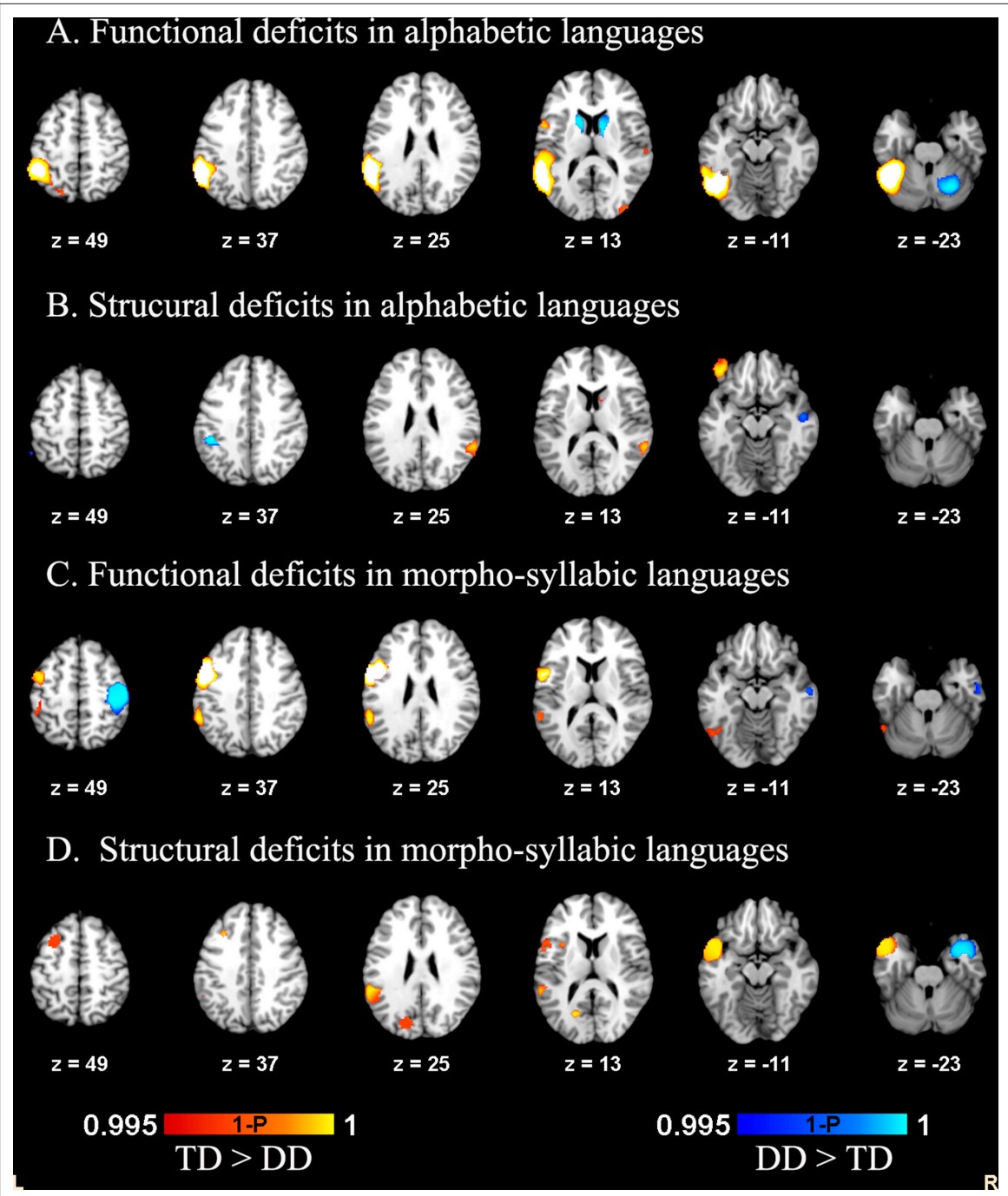

**Figure 1.** Functional and structural deficits related to DD in alphabetic languages and morpho-syllabic languages.

at right STG (*Table 1* and *Figure 1A*). Hyperactivation in DD was found in the right cerebellum and bilateral caudate nucleus.

In the meta-analysis of functional studies in morpho-syllabic languages, hypoactivation in DD was found in left IFG opercular part, left supramarginal gyrus and left ITG. Hyperactivation in DD was found in right precentral gyrus and right middle temporal gyrus (MTG) (*Table 2*, *Table 3* and *Figure 1C*). The jack-knife sensitivity analysis showed that all results reported above were replicable (*Table 1*; *Table 3*).

**Table 2.** Structural deficits in individuals with DD in alphabetic languages (JK represents the results of jack-knife sensitivity analysis).

| Regions | MNI coordinate | SDM-Z | p | Voxels | Cluster breakdown (Voxels) | JK |
|---|---|---|---|---|---|---|
| *Decreased GMV in DD* | | | | | | |
| Left IFG orbital part | −38,42,−16 | 2.306 | 0.0001 | 611 | Left IFG orbital part, BA 47 (217) | 20/21 |
| Right STG | 56,−44,18 | 2.024 | 0.0003 | 560 | Right STG, BA 42 (156) | 20/21 |
| Right caudate | 6,14,2 | 1.695 | 0.0022 | 166 | | 21/21 |
| *Increased GMV in DD* | | | | | | |
| Left IPL | −42,−36,36 | −1.976 | 0.0000 | 237 | Left IPL, BA 40 (237) | 20/21 |
| Right MTG | 50,−12,−14 | −1.040 | 0.0014 | 174 | | 21/21 |

## Structural deficits in alphabetic languages and morpho-syllabic languages

In the meta-analysis of structural studies in alphabetic languages, readers with DD showed a decrease in GMV in the left IFG orbital part, right STG and right caudate nucleus (*Table 2* and *Figure 1B*). In contrast, readers with DD showed an increase in GMV in the left IPL and right MTG. In the meta-analysis of structural studies in morpho-syllabic languages, readers with DD showed a decrease in GMV in the left temporoparietal cortex, left calcarine cortex and left MFG. Readers with DD showed an increase in GMV in the right STG (*Table 4* and *Figure 1D*). The jack-knife sensitivity analysis showed that all results reported were replicable (*Table 2*; *Table 4*).

In the supplementary materials, we reported whether the structural and functional deficits found in the current study were reported in each study included in the meta-analysis (*Supplementary file 1f-1i*).

## Comparison between alphabetic and morpho-syllabic languages

For the direct comparison between the morpho-syllabic and alphabetic groups in functional studies, we found greater reduction of brain activation in alphabetic languages than in morpho-syllabic languages in the left MTG, right STG and left fusiform gyrus. We found greater reduction of brain activation in morpho-syllabic languages than in alphabetic languages in the left IFG, opercular part and greater increase of brain activation in DD in morpho-syllabic languages than in alphabetic languages in the right precentral gyrus (*Table 5*, *Figure 2A*).

**Table 3.** Functional deficits in individuals with DD in morpho-syllabic languages (JK represents the results of jack-knife sensitivity analysis).

| Regions | MNI coordinate | SDM-Z | p | Voxels | Cluster breakdown (Voxels) | JK |
|---|---|---|---|---|---|---|
| *Hypoactivation in DD* | | | | | | |
| Left IFG opercular part | −48,10,28 | 4.071 | 0.0000 | 2527 | Left precentral gyrus, BA 6 (623)<br>Left IFG opercular part, BA 44 (278)<br>Left precentral gyrus, BA 44 (195)<br>Corpus callosum (178)<br>Left MFG, BA 44 (162) | 12/12 |
| Left supramarginal gyrus | −58,−42,26 | 2.149 | 0.0001 | 1001 | Left IPL, BA 40 (271)<br>Left STG, BA 42 (153)<br>Left supramarginal gyrus, BA 48 (144) | 11/12 |
| Left ITG | −48,−56,−18 | 1.761 | 0.0008 | 326 | Left ITG, BA37 (166) | 9/12 |
| *Hyperactivation in DD* | | | | | | |
| Right precentral gyrus | 52,−16,44 | −2.035 | 0.0000 | 2201 | Right precentral gyrus, BA 6 (640)<br>Right postcentral gyrus, BA 3 (447)<br>Right precentral gyrus, BA 4 (350)<br>Right postcentral gyrus, BA 4 (215) | 12/12 |
| Right MTG | 56,−10,−18 | −1.453 | 0.0013 | 298 | | 10/12 |

**Table 4.** Structural deficits in individuals with DD in morpho-syllabic languages (JK represents the results of jack-knife sensitivity analysis).

| Regions | MNI coordinate | SDM-Z | p | Voxels | Cluster breakdown (Voxels) | JK |
|---|---|---|---|---|---|---|
| *Decreased GMV in DD* | | | | | | |
| Left STG | −50,4,−4 | 2.466 | 0.0000 | 2,948 | Left insula, BA 48 (539)<br>Left STG, BA 38 (392)<br>Left rolandic operculum, BA 48 (226)<br>Left MTG, BA 21 (215)<br>Left STG, BA 48 (186) | 6/6 |
| Left temporoparietal cortex | −56,−40,18 | 2.102 | 0.0002 | 900 | Left supramarginal gyrus, BA 48 (188)<br>Left STG, BA 42 (171) | 6/6 |
| Left calcarine cortex | −20,−66,14 | 2.447 | 0.0000 | 449 | Corpus callosum (297) | 6/6 |
| Left MFG | −32,26,40 | 2.319 | 0.0001 | 438 | | 6/6 |
| *Increased GMV in DD* | | | | | | |
| Right STG | 34,6,−26 | −1.572 | 0.0001 | 1,829 | Right STG, BA 38 (261)<br>Right ITG, BA 20 (250)<br>Right MTG, BA 20 (212) | 6/6 |
| Right precuneus | 12,−52,42 | −1.254 | 0.0014 | 156 | | 2/6 |

**Table 5.** Direct comparisons between alphabetic languages and morpho-syllabic languages in functional studies.

| Regions | MNI coordinate | SDM-Z | p | Voxels | Cluster breakdown (Voxels) |
|---|---|---|---|---|---|
| *Hypoactivation in DD* | | | | | |
| *Alphabetic languages > Morpho-syllabic languages* | | | | | |
| Left MTG | −54,−62,8 | 1.203 | 0.0000 | 2,173 | Left MTG, BA 37 (453)<br>Left MTG, BA 48 (294)<br>Left MTG, BA 21 (293)<br>Corpus callosum (184) |
| Right STG | 60,−18,4 | 1.080 | 0.0000 | 2047 | Right STG, BA 22 (400)<br>Corpus callosum (373)<br>Right insula, BA 48 (278)<br>Right STG, BA 48 (259)<br>Right rolandic operculum, BA 48 (193) |
| Left fusiform gyrus | −40,−42,−24 | 1.279 | 0.0000 | 924 | Left fusiform gyrus, BA 37 (198)<br>Left ITG, BA 20 (198) |
| *Morpho-syllabic languages > Alphabetic languages* | | | | | |
| Left IFG opercular part | −48,8,30 | −3.945 | 0.0000 | 2093 | Left precentral gyrus, BA 6 (512)<br>Left IFG opercular part, BA 44 (274)<br>Left precentral gyrus, BA 44 (191)<br>Corpus callosum (161)<br>Left IFG, triangular part, BA 48 (159) |
| *Hyperactivation in DD* | | | | | |
| *Morpho-syllabic languages > Alphabetic languages* | | | | | |
| - | - | - | - | - | - |
| *Morpho-syllabic languages > Alphabetic languages* | | | | | |
| Right precentral gyrus | 40,−20,54 | −2.262 | 0.0000 | 1,518 | Right precentral gyrus, BA 6 (525)<br>Right precentral gyrus, BA 4 (306)<br>Right postcentral gyrus, BA 3 (286)<br>Right postcentral gyrus, BA 4 (171) |

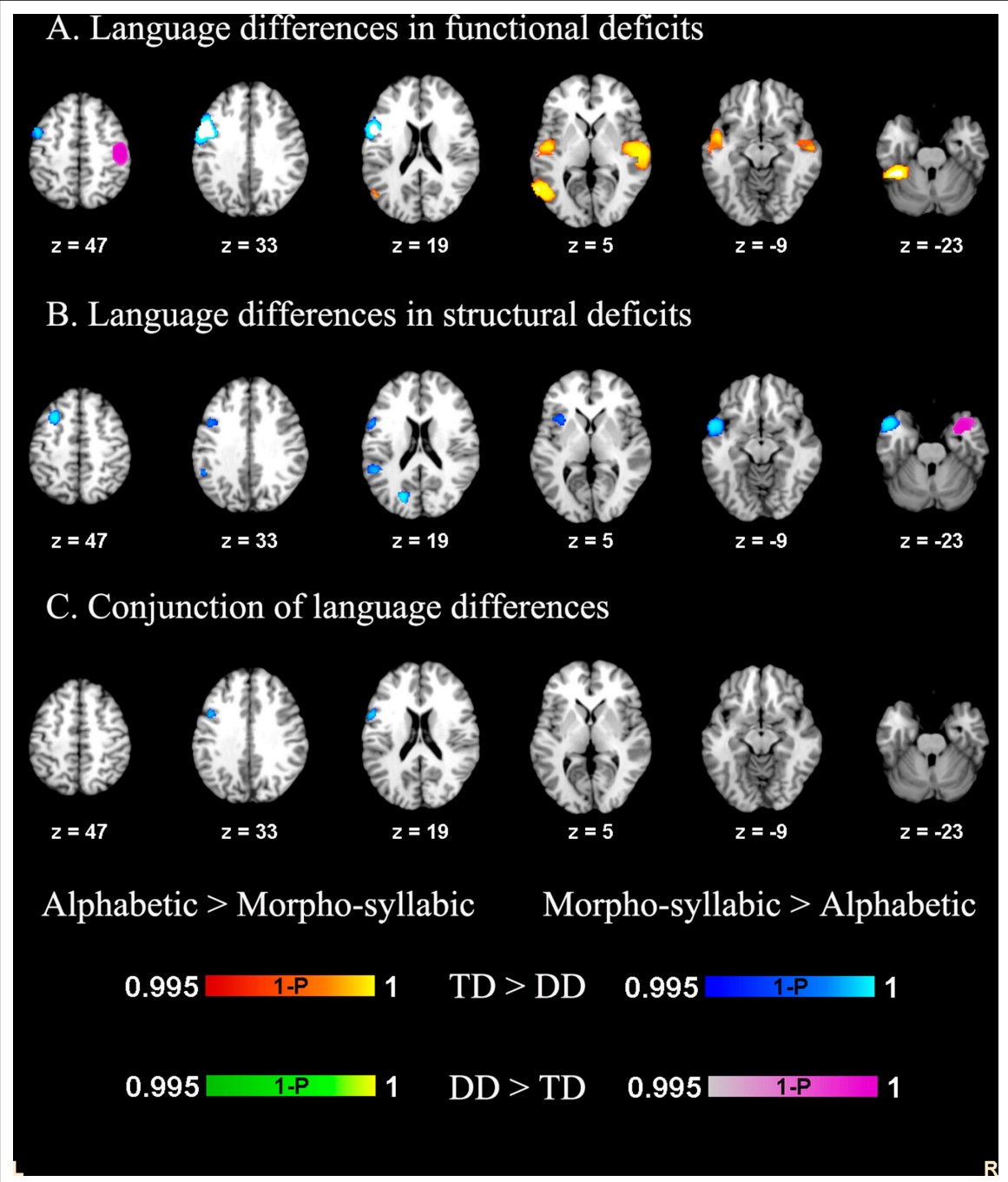

**Figure 2.** Direct comparisons between the alphabetic group and the morpho-syllabic group in structural and functional deficits. Conjunction analysis showed greater reduction of both GMV and brain activation in the left dorsal IFG in morpho-syllabic languages than alphabetic languages.

The online version of this article includes the following figure supplement(s) for figure 2:

**Figure supplement 1.** Language differences in functional studies and structural studies in children.

**Figure supplement 2.** Functional deficits in children with DD in each group and common deficits between them.

**Figure supplement 3.** Language differences between the two well-matched groups in the confirmation analysis and how the current results overlap with the original results.

For the direct comparison between the morpho-syllabic and alphabetic groups in structural studies, we found greater reduction of GMV in DD in morpho-syllabic languages than in alphabetic languages in the left STG, left IFG opercular part, left MFG, left supramarginal gyrus, left superior occipital gyrus

**Table 6.** Direct comparisons between alphabetic languages and morpho-syllabic languages in structural studies.

| Regions | MNI coordinate | SDM-Z | P | Voxels | Cluster breakdown (Voxels) |
|---|---|---|---|---|---|
| *Decreased GMV in DD* | | | | | |
| *Alphabetic languages > Morpho-syllabic languages* | | | | | |
| - | - | - | - | - | - |
| *Morpho-syllabic languages > Alphabetic languages* | | | | | |
| Left STG | −50,10,−18 | −3.139 | 0.0000 | 1,218 | Left STG, BA 38 (322) |
| Left IFG opercular part | −52,10,24 | −2.253 | 0.0008 | 409 | |
| Left MFG | −28,16,44 | −2.888 | 0.0001 | 346 | |
| Left supramarginal gyrus | −48,−44,24 | −2.761 | 0.0001 | 344 | |
| Left SOG | −20,−72,20 | −2.911 | 0.0001 | 277 | Corpus callosum (187) |
| Left insula | −32,14,8 | −2.046 | 0.0015 | 179 | |
| *Increased GMV in DD* | | | | | |
| *Alphabetic languages > Morpho-syllabic languages* | | | | | |
| - | - | - | - | - | - |
| *Morpho-syllabic languages > Alphabetic languages* | | | | | |
| Right STG | 34,6,−28 | −1.893 | 0.0001 | 1,397 | Right STG, BA 38 (246) |
| Left ITG | −54,−58,−16 | −1.112 | 0.0018 | 161 | |

(SOG) and left insula. We also found greater increase of GMV in DD in morpho-syllabic languages than in alphabetic languages in the right STG and left ITG (*Table 6*, *Figure 2B*). We found no regions that showed greater GMV alterations in alphabetic languages than in morpho-syllabic languages.

To identify the common language differences between the structural and functional studies, we conducted a conjunction analysis between the language differences in structural studies and functional studies. This produced an overlap of 377 voxels in the left IFG opercular part, with a peak at (–52, 10, 24), indicating greater reduction of both GMV and brain activation in morpho-syllabic languages than in alphabetic languages (*Figure 2C*).

## Multimodal analysis results in alphabetic and morpho-syllabic languages

Multimodal meta-analysis in alphabetic languages showed that decreased GMV and hypoactivation in DD were found in the bilateral STG and left IFG triangular part; no regions showed increased GMV and hyperactivation; increased GMV and hypoactivation in DD were found in left IPL and left cerebellum; decreased GMV and hyperactivation in DD were found in bilateral caudate and right cerebellum (*Table 7*, *Figure 3A*). Multimodal meta-analysis in morpho-syllabic languages showed that decreased GMV and hypoactivation in DD were found in the left STG and left IFG opercular part; increased GMV and hyperactivation in DD were found in the right MTG; decreased GMV and hyperactivation in DD were found in left STG; no regions showed increased GMV and hypoactivation (*Table 8*, *Figure 3B*).

To identify the common multimodal deficits in alphabetic languages and morpho-syllabic languages, we conducted a conjunction analysis of the thresholded multimodal maps of the two types of writing systems. This procedure produced an overlap of 482 voxels in the left STG, which peaked at (−54,−34, 20), indicating shared reduction of GMV and hypoactivation in both types of writing systems (*Figure 3C*).

## Confirmation analysis results

For the confirmation analysis on children only in the alphabetic group, when we compared children in alphabetic languages to children in morpho-syllabic languages, we found a conjunction of the language differences in the functional studies and the structural studies, which was greater reduction

**Table 7.** Multimodal structural and functional abnormalities in individuals with DD in alphabetic languages.

| Regions | MNI coordinate | Voxels | Cluster breakdown (Voxels) |
|---|---|---|---|
| *Decreases of GMV and hypoactivation in DD* | | | |
| Left STG | −52,−30,20 | 1,099 | Left MTG, BA 21 (237) |
| Right STG | 62,−32,14 | 322 | |
| Left IFG, triangular part | −46,42,0 | 219 | |
| *Increases of GMV and hypoactivation in DD* | | | |
| Left IPL | −46,−40,38 | 1,689 | Left IPL, BA 40 (941) |
| Left cerebellum | −40,−70,−24 | 446 | Left cerebellum, crus I, BA 19 (159) |
| *Decreases of GMV and hyperactivation in DD* | | | |
| Right cerebellum | 28,−52,−34 | 1,286 | Right cerebellum, lobule VI, BA 37 (273)<br>Middle cerebellar peduncles (267)<br>Right cerebellum, lobule VI, BA 19 (204) |
| Right caudate | 8,8,12 | 600 | Right anterior thalamic projections (214)<br>Right caudate nucleus (189) |
| Left caudate | −16,8,14 | 595 | Left anterior thalamic projections (353) |

of brain activation and GMV in morpho-syllabic languages than in alphabetic languages in the left IFG opercular part (–54, 10, 20) with a cluster of 273 voxels (*Figure 2—figure supplement 1C*), which is consistent with the original result. Conjunction analysis of the functional alterations in the alphabetic group and the morpho-syllabic group revealed common reduction of brain activation in children with DD in the left ITG (−48,−56, −18) and the left TP area (−56,−44, 32) with a cluster size of 291 voxels and 778 voxels, respectively. The TP area overlapped with the original multimodal result at the left STG (−56,−48, 22) where both language groups showed reduced brain activation and GMV. Taken together, these results are consistent with the original results, suggesting that the language differences are not due to unmatched age range. For detailed results of the confirmation analysis, please see *Supplementary file 1a-1d*, *Figure 2—figure supplement 1*, *Figure 2—figure supplement 2*.

Furthermore, the other confirmation analysis on functional studies of two well-matched subgroups on age, task and number of studies also confirmed our original findings. Conjunction analysis between the original results and the current functional differences between English and Chinese showed consistent greater reduction of brain activation in DD in morpho-syllabic languages/Chinese than in alphabetic languages/English in the left dorsal IFG (–52, 6, 16) with a cluster size of 1966 voxels. There was consistent greater reduction of brain activation in DD in alphabetic languages/English than in morpho-syllabic languages/Chinese in the left MTG (−48,−68, 4) and left MOG (−51,−41, −24) with a cluster size of 166 voxels and 90 voxels, respectively. Consistent greater increase of brain activation in morpho-syllabic languages/Chinese than in alphabetic languages/English was also found in the right precentral gyrus (44, -22, 50), and the cluster size was 792 voxels (*Figure 2—figure supplement 3C*). For detailed results of this confirmation analysis, please see *Supplementary file 1e*, *Figure 2—figure supplement 3*.

## Discussion

In this meta-analysis study, we examined the convergence and divergence between the brain structural and functional deficits associated with DD as well as whether the deficits are consistent across languages. We found that readers with DD showed both GMV reduction and functional hypoactivation in the left TP and ventral IFG in alphabetic languages, readers with DD showed both GMV reduction and functional hypoactivation in the left TP and dorsal IFG in morpho-syllabic languages,

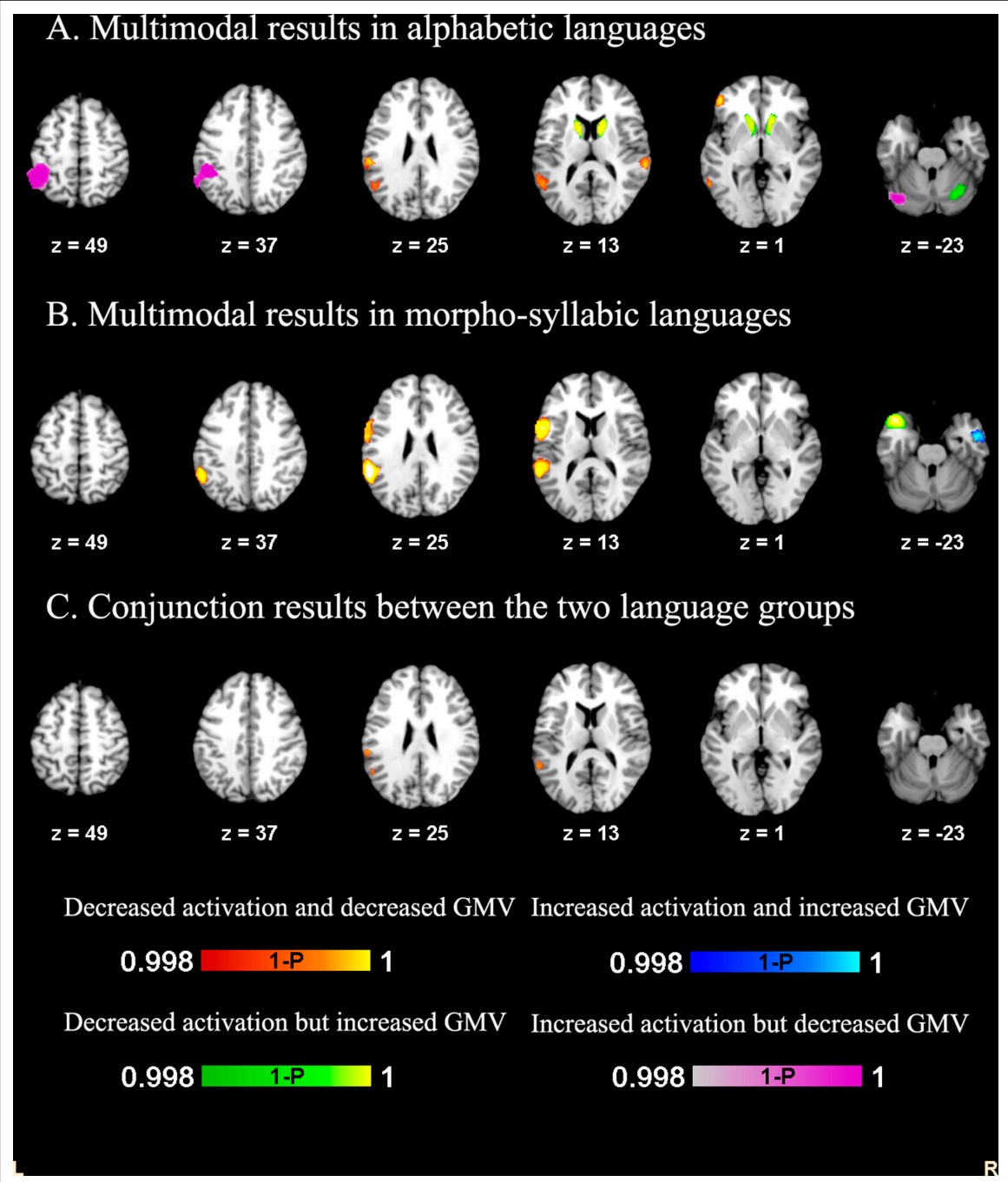

**Figure 3.** Structural and functional deficits in DD for alphabetic languages and morpho-syllabic languages. Decreased GMV and brain activation were found in both groups in the left STG.

among which, the left STG was a shared impairment across all languages, and the dorsal left IFG showed a greater impairment in morpho-syllabic languages than in alphabetic languages, suggesting both language-universal and language-specific deficits in the brain. We also found GMV increase and functional hyperactivation in the right anterior MTG/ITG region in morpho-syllabic languages; however, conjunction analysis between morpho-syllabic languages and alphabetic languages did not reveal any overlap. In addition to the consistent structural and functional alterations, we also detected inconsistent structural and functional alterations. Individuals with DD showed increased GMV and hypoactivation in the left IPL and left cerebellum, and decreased GMV and hyperactivation in the bilateral caudate in alphabetic languages, but decreased GMV and hyperactivation in left STG in

**Table 8.** Multimodal structural and functional abnormalities in individuals with DD in morpho-syllabic languages.

| Regions | MNI coordinate | Voxels | Cluster breakdown (Voxels) |
|---|---|---|---|
| *Decreases of GMV and hypoactivation in DD* | | | |
| Left STG | −58,−38,22 | 1,566 | Left supramarginal gyrus, BA 48 (254)<br>Left STG, BA 42 (253)<br>Left supramarginal gyrus, BA 40 (153) |
| Left IFG opercular part | −56,2,10 | 1,052 | Left IFG opercular part, BA 44 (151) |
| *Decreases of GMV and hyperactivation in DD* | | | |
| Left STG | −44,16,−22 | 854 | Left STG, BA 38 (394) |
| *Increases of GMV and hyperactivation in DD* | | | |
| Right MTG | 48,−6,−26 | 493 | Right MTG, BA 21 (178) |

morpho-syllabic languages. However, conjunction analysis between morpho-syllabic languages and alphabetic languages did not reveal any overlap.

## Convergent structural and functional impairment across writing systems

Across writing systems, convergent structural and functional deficit was found in the left STG due to reduced GMV and brain activation in both alphabetic languages and morpho-syllabic languages. Further confirmation analysis confirmed that this is a stable deficit across children and adults. This is consistent with previous meta-analysis studies (*Maisog et al., 2008*; *McGrath and Stoodley, 2019*; *Paulesu et al., 2014*; *Richlan et al., 2013*). The left STG is a very important component in the language network (*Friederici, 2012*; *Hickok and Poeppel, 2007*) as well a key region in the reading network (*Pugh et al., 2000*; *Richlan, 2014*). It is involved in phonological representation and phonological processing during both spoken language processing and reading (*Bolger et al., 2005*; *Enge et al., 2020*; *Tan et al., 2005*). Recent research has suggested that proficient reading is characterized by convergence between speech and print at this region regardless of languages, as multivariate brain activity patterns are similar for speech and print at this region (*Chyl et al., 2021*). Therefore, the reading network may develop based on the built-in language circuit, as reading is a skill that humans acquire too late in the course of human evolution to have a brain network dedicated to it. Recently, a growing number of studies have investigated early signs of dyslexia before the onset of reading and found that structural and functional deficits in the left TP area and left inferior frontal cortex appear before reading onset (*Clark et al., 2014*; *Hosseini et al., 2013*; *Plewko et al., 2018*; *Raschle et al., 2012*; *Raschle et al., 2014*; *Vandermosten et al., 2019*). It further suggests that DD might be due to early abnormality in the language network. Specifically, *Skeide et al., 2018* found hypermyelination in the left auditory cortex in readers with DD using ultra-high-field MRI at 7T, and disrupted neural firing induced by hypermyelination in the layer IV of the auditory cortex, which may cause hypoactivation in the left STG. The left STG actually serves as an important hub in the language and reading network (*Fernández et al., 2020*), which connects the inferior frontal network and the OT network through a dorsal pathway and a ventral pathway (*Brauer et al., 2013*; *Cummine et al., 2015*). Disconnection with the left STG has been verified in task-related functional connectivity studies (*Boets et al., 2013*; *Cao et al., 2017*; *Schurz et al., 2015*) and a resting-state functional connectivity study (*Schurz et al., 2015*), as well as in a meta-analysis of DTI studies (*Vandermosten et al., 2012*). Our finding suggests that dyslexia is associated with structural and functional abnormalities of the left STG regardless of language. The evidence suggests that this is a neural signature of DD, which supports the phonological deficit hypothesis (*Shaywitz et al., 1998*).

However, we failed to find consistent structural and functional deficits in the OT area. The main reason was that there was no structural alteration but only functional reduction at this area. The OT area is a key region for orthographic recognition during visual word processing (*Glezer et al., 2009*; *Glezer et al., 2016*; *Hirshorn et al., 2016*; *Nobre et al., 1994*) and was reported to be impaired in

individuals with DD (*McCrory et al., 2005*; *Richlan et al., 2010*; *Wandell et al., 2012*). The specialization of this region for orthographic processing is developed along with reading acquisition (*Brem et al., 2010*), and the dysfunction of the OT area in DD is possibly a result of reading failure (*Pugh et al., 2000*). A recent meta-analysis of VBM studies (*McGrath and Stoodley, 2019*) also failed to detect structural deficit in the OT area, which is consistent with our finding. Taken together, the lack of structural deficits with only hypoactivation at the OT area appears to suggest that the visuo-orthographic deficits at the OT might be a consequence of being DD. In contrast, the left STG which was discussed above, appears to be associated with the cause of DD. Our results provide further support for the phonological deficit hypothesis that phonological deficit is the primary deficit and other deficits may be a result of the phonological deficit (*Pugh et al., 2000*).

## Language differences in structural and functional alterations

The left dorsal IFG which peaked at (–52, 10, 24) showed greater reduction in both GMV and brain activation in morpho-syllabic languages than in alphabetic languages, suggesting greater impairment in this region in morpho-syllabic languages than in alphabetic languages. This finding is also verified in the confirmation analyses. Previously, many Chinese studies have reported impairment at the dorsal left IFG, for example, reduced brain activation in an auditory rhyming judgment task in children with DD at (–44, 10, 26) (*Cao et al., 2017*), in a lexical decision task at (–44, 3, 29) (*Siok et al., 2004*), in a homophone judgment task at (–55, 5, 22) (*Siok et al., 2008*), and in a morphological task at (–36, 8, 26) (*Liu et al., 2013a*). This left dorsal IFG has been believed to be more involved in Chinese reading than in alphabetic languages, with a peak at the left MFG (–46, 18, 28) as reported in a previous meta-analysis study (*Tan et al., 2005*). The dorsal IFG was found to be involved in phonological processing in Chinese reading (*Wu et al., 2012*), and it is thought to be related to addressed phonology during Chinese character reading (*Tan et al., 2005*). Our study adds to the literature that by direct comparison, this region does show greater deficit in individuals with DD in morpho-syllabic languages than in alphabetic languages in terms of both brain activation and GMV. This might be due to the fact that healthy Chinese readers have increased GMV and brain activation in the left dorsal IFG than healthy alphabetic readers, because the features of Chinese require greater involvement of this region in reading than alphabetic languages due to the whole-character-to-whole-syllable mapping. Actually, two cross-linguistic studies have argued that different findings of DD in different languages are actually driven by the fact that control readers show language-specific brain activation patterns (*Feng et al., 2020*; *Hu et al., 2010*), and that brain activation in individuals with DD is actually the same across languages. For example, *Hu et al., 2010* found that Chinese control readers showed greater activation in the left IFG, and English control readers showed greater activation in the left superior temporal sulcus; however, children with DD in Chinese and English showed similar brain activation in these two regions. Therefore, readers with DD fail to show language specialization due to their limited reading experience and skills. In summary, this language-specific deficit is believed to be a consequence of being DD in learning morpho-syllabic languages, indicating their inability to accommodate to their own writing system.

In the direct comparison between alphabetic and morpho-syllabic languages, we also found greater hypoactivation in DD in alphabetic languages than in morpho-syllabic languages in the left MTG, right STG and left fusiform gyrus, which was verified in the well-matched confirmation analysis, suggesting that the language difference should not be due to differences in age, tasks, and number of studies in the two language groups. Our finding is consistent with previous neuroimaging studies that revealed reduced activation associated with DD in the posterior reading network in alphabetic languages (*Paulesu et al., 2001*; *Richlan et al., 2010*; *Vandermosten et al., 2019*), suggesting deficient orthographic and phonological processing. However, the novelty of the current study is to demonstrate greater severity of deficit in these regions in alphabetic languages than in morpho-syllabic languages. In a previous meta-analysis study of alphabetic languages, it was found that there was greater hypoactivation in the left fusiform gyrus (–40,–42, –16) in shallow orthographies than in deep orthographies (*Martin et al., 2016*). The explanation is that this region is associated with bottom-up rapid processing of letters, because it was found that at a proximal region (–38,–50, –16), there was a word length effect for German nonwords in non-impaired readers (*Schurz et al., 2010*). Moreover, the left fusiform gyrus has been found to be more involved in English reading than in Chinese reading in typical mature readers with a peak of the effect at (–44,–56, –12) (*Tan et al., 2005*). Therefore, the

left fusiform gyrus is important for letter-by-letter orthographic recognition in alphabetic languages, and this explains why we found greater deficit in this region in alphabetic languages than in morpho-syllabic languages. As for the right STG, the previous meta-analysis found greater hypoactivation in deep orthographies than shallow orthographies (*Martin et al., 2016*). Together with our finding, it suggests that the right STG might be associated with the inconsistent mapping between graphemes and phonemes in deep orthographic alphabetic languages. In summary, these greater deficits in alphabetic languages than in morpho-syllabic languages might be due to the inability to adapt to the special features of alphabetic languages in individuals with DD.

For the structural studies, we found greater GMV alterations in morpho-syllabic languages than in alphabetic languages, including greater GMV reduction in the left STG, left MFG, left supramarginal gyrus, left SOG and left insula, as well as greater GMV increase in the right STG and ITG. However, considering the limited number of studies included in the morpho-syllabic language group and inconsistent results with functional studies, the results should be interpreted with caution.

## Increased GMV and hyperactivation

In the multi-modal meta-analysis, we found increased GMV and hyperactivation in participants with DD in the right MTG which was driven by the morpho-syllabic languages. For the functional studies, we also found greater hyperactivation in the right precentral gyrus in morpho-syllabic languages than in alphabetic languages. These alterations might be related to the compensation mechanism of the right hemisphere. As precentral gyri play an important role in articulation (*Dronkers, 1996*), overactivation in the precentral gyrus is interpreted as an articulation strategy used by individuals with DD to compensate for their deficient phonological processing (*Cao et al., 2018*; *Shaywitz et al., 1998*; *Waldie et al., 2013*). The compensation in the right MTG is developed in morpho-syllabic languages presumably due to the tight connection between orthography and semantics (*Wang et al., 2015*). Substantial evidence has shown that dyslexia was often accompanied by excessive activation of the right hemisphere (*Cao et al., 2017*; *Cao et al., 2018*; *Kovelman et al., 2012*; *Kronschnabel et al., 2014*; *Yang and Tan, 2020*) and reduced left lateralization of the language network (*Altarelli et al., 2014*; *Bloom et al., 2013*). Furthermore, training studies have also found increased activation in many regions in the right hemisphere in individuals with DD after reading intervention (*Barquero et al., 2014*; *Meyler et al., 2008*), suggesting the compensatory role of the right hemisphere when the left language/reading network is deficient (*Coslett and Monsul, 1994*; *Weiller et al., 1995*). However, according to the previous meta-analysis study, different regions showed overactivation in different writing systems (*Martin et al., 2016*). In particular, the left anterior insula showed greater overactivation in deep orthographies while the left precentral gyrus showed greater overactivation in shallow orthographies in individuals with DD. Taken together, it suggests that different compensatory mechanisms are developed depending on the characteristics of the writing system as well as learning experiences, and the compensation in the right MTG and right precentral gyrus appears to be particularly salient in morpho-syllabic languages. However, it is also possible that the increased GMV and hyperactivation in individuals with DD are due to some fundamental deficits rather than compensation. Further research is needed to understand the nature of these alterations by running brain-behavioral correlation and/or employing longitudinal designs.

## Divergent structural and functional alterations in DD

In the multimodal analysis, we also found divergent structural and functional changes related to DD, including the left IPL and left cerebellum where there was increased GMV and hypoactivation and bilateral caudate where there was reduced GMV and hyperactivation in alphabetic languages; There was decreased GMV and hyperactivation in left STG in morpho-syllabic languages. This is in line with a recent study which found a dissociation between the developmental changes of brain structure and function (*Siok et al., 2020*), suggesting that learning experience may sometimes shape the brain function independent of the brain structure.

## The left IPL

We found increased GMV and hypoactivation in DD in the left IPL in alphabetic languages. Consistent hypoactivation in the left IPL in DD has been documented in previous studies (*Maisog et al., 2008*; *Martin et al., 2016*). Furthermore, it was found that the deficit of the left IPL was greater in children

than in adults with DD (**Richlan et al., 2011**), suggesting that the functional impairment of the left IPL may gradually recover with development. This may be related to the transfer of the reading circuit from the dorsal pathway to the ventral pathway over development (**Younger et al., 2017**). Control children activate the left IPL to a greater degree than control adults because they rely more on the dorsal pathway. Therefore, children with DD show a great reduction in the left IPL in comparison to adults with DD. Alternatively, the left IPL has been found to be deactivated during language tasks (**Cao et al., 2008**; **Cao et al., 2017**; **Meyler et al., 2008**; **Schulz et al., 2009**), and this is due to the nature of the default mode network (**Laird et al., 2009**), which is deactivated during active tasks. Therefore, it might be the case that the increased GMV in individuals with DD increases inhibitory inputs received by the IPL, which results in greater deactivation.

For structural studies involving the left IPL, the results are inconsistent. The GMV of the left supramarginal gyrus around the posterior part of perisylvian cortex was found to be reduced in individuals with DD (**Linkersdörfer et al., 2012**; **McGrath and Stoodley, 2019**) and it showed a positive correlation with reading accuracy only in normal readers (**Jednorog et al., 2015**). However, the GMV of the left inferior parietal cortex excluding the supramarginal and the angular was found to increase in individuals with DD (**McGrath and Stoodley, 2019**) and a study showed that the volume of the left inferior parietal cortex in control readers was negatively correlated with reading level (**Houston et al., 2014**). The IPL in the current study is outside the supramarginal and angular gyrus; therefore, it is consistent with the previous findings that there is increased GMV in individuals with DD.

## The cerebella

Increased GMV and hypoactivation in DD were also found in the left cerebellum in alphabetic languages; however, in the right cerebellum, we found decreased GMV and hyperactivation. Previously, it was found that the right cerebellum is greater in size than the left cerebellum in healthy controls while the asymmetry is reduced in individuals with DD (**Kibby et al., 2008**; **Rae et al., 2002**). This is consistent with our finding of increased GMV in the left cerebellum and decreased GMV in the right cerebellum in individuals with DD, suggesting reduced asymmetry in cerebellum.

The cerebella have been found to play an important role in inner speech, automatization in reading and suppression of overt articulatory movement in silent reading (**Ait Khelifa-Gallois et al., 2015**). Functional abnormality of cerebellum in DD has been reported repeatedly; however, hyperactivation was reported more often in the right cerebellum (**Feng et al., 2017**; **Hernandez et al., 2013**; **Kronschnabel et al., 2014**; **Richlan et al., 2010**; **Rumsey et al., 1997**; **van Erminen-Marbach et al., 2013a**), while hypoactivation was reported more often in the left cerebellum (**Christodoulou et al., 2014**; **McCrory et al., 2000**; **Olulade et al., 2012**; **Reilhac et al., 2013**; **Siok et al., 2008**). This is consistent with our finding of hyperactivation in the right cerebellum and hypoactivation in the left cerebellum. Different alteration patterns in the left and right cerebellum suggest that they may play different roles in reading and dyslexia. The right cerebellum has been found to be connected with the left frontal-parietal pathway for phonological processing and with the left frontal-temporal pathway for semantic processing (**Alvarez and Fiez, 2018**; **Gatti et al., 2020**). The left cerebellum, however, is involved in error monitoring during reading unfamiliar non-words (**Ben-Yehudah and Fiez, 2008**), as well as articulation related movement process, since it is activated in reading aloud but not in lexical decision (**Carreiras et al., 2007**). **Richards et al., 2006** argued that the left cerebellum is involved in processing the morphology of word forms, and the right cerebellum is involved in phonological processing. Therefore, hyperactivation in the right cerebellum in readers with DD suggests that they may use it as a compensation for their deficient phonological processing, while hypoactivation in the left cerebellum may suggest reduced error monitoring in readers with DD. The finding of neurological alterations in the cerebellum is consistent with the previous findings of cerebellar deficit (**Menghini et al., 2006**; **Yang et al., 2013**) for which, some researchers argued that impaired articulatory motor control in the cerebellum leads to reading impairment (**Nicolson et al., 2001**; **Stoodley and Stein, 2011**). It also should be noticed that the cerebellum showed divergent patterns in structural studies and functional studies. Taken together, these results implicate the necessity of considering DD from a broader spectrum of developmental disorders.

It is still unclear why there is increased GMV but decreased activation in some brain regions. It may be due to the following reasons: (1) increased dendrites receiving more inhibitory input from other neurons; (2) abnormal neuronal migration deactivated the firing of neurons as a result of disrupted

local microcircuits (*Giraud and Ramus, 2013*); (3) weaker input from other regions deactivated the target region and changed the structure of the region (*Wang et al., 2019*).

## The caudate

We also found decreased GMV and hyperactivation in readers with DD in the bilateral caudate in alphabetic languages. Previous studies have observed decreased GMV (*Brown et al., 2001*; *Jagger-Rickels et al., 2018*; *McGrath and Stoodley, 2019*; *Tamboer et al., 2015*) and hyperactivation in bilateral caudate in individuals with DD (*Martin et al., 2016*; *Olulade et al., 2012*; *Pekkola et al., 2006*; *Richlan et al., 2010*; *Richlan et al., 2011*; *Rumsey et al., 1997*). However, there are also studies that observed different patterns (*Cheema et al., 2018*), such as decreased activation in caudate (*Perrachione et al., 2016*). Furthermore, the GMV volume of the caudate in individuals with DD was found to be positively correlated with reading performance (*Pernet et al., 2009*; *Tamboer et al., 2015*), and the left caudate's activation was correlated with longer reaction time in word reading only in individuals with DD (*Cheema et al., 2018*). The caudate plays an important role in procedural learning and phonological processing (*Grahn et al., 2008*; *Tettamanti et al., 2005*; *Ullman et al., 2020*). Decreased GMV and increased activation at the bilateral caudate might be caused by reduced dendrites and reduced inhibitory inputs received in individuals with DD (*Achal et al., 2016*; *Finn et al., 2014*). It may also be due to pre-existing local structural deficit leading to compensatory hyperactivation of the remaining part of the caudate. GMV reduction in basal ganglia was found in many other neuropsychiatric disorders, such as attention-deficit hyperactivity disorder (*Frodl and Skokauskas, 2012*; *Mous et al., 2015*; *Nakao et al., 2011*), autism spectrum disorder (*Nickl-Jockschat et al., 2012*) and major depression disorder (*Husain et al., 1991*; *Lu et al., 2016*). Altered myelination and neurotransmitters may contribute to the structural and functional alterations related to basal ganglia (*Nord et al., 2019*; *Wichmann and DeLong, 2012*).

## Conclusion

We found convergent functional and structural alterations in the left STG across different writing systems, suggesting a neural signature of DD, which might be associated with phonological deficit. We also found greater functional and structural alteration in the left dorsal IFG in morpho-syllabic languages than alphabetic languages, suggesting a language-specific effect of DD, which might be related to the special feature of whole-character-to-whole-syllable mapping in morpho-syllabic languages.

## Limitation

In this meta-analysis, we found convergent and divergent functional and structural alterations across writing systems. However, due to the limitations of voxel-based neuroimaging meta-analysis, the peak coordinate only provides limited information, therefore, future image-based meta-analysis studies should be conducted with full statistical images of the original studies (*Muller et al., 2018*).

# Materials and methods

## Literature retrieval and data extraction

We searched in 'PubMed' (http://www.pubmed.org) and 'Web of science' for neuroimaging studies published from January 1986 to January 2020 using a combination of a condition term (i.e. dyslexia, reading disorder, reading impairment, reading difficulty or reading disability) and a technical term (i.e. fMRI, PET, voxel-based morphometry, VBM, or neuroimaging), for example, 'dyslexia' and 'fMRI'. See the full list of key word combinations in the *Supplementary file 2*. Additionally, we manually added studies by checking the references of the selected papers that were missed in the search. The inclusion criteria were: (1) PET, fMRI, voxel-based morphometry (VBM) studies or structural studies using a volumetric FreeSurfer pipeline, (2) whole-brain results were reported, (3) direct group comparisons between readers with DD and age control readers were reported, (4) coordinates were reported in Talairach or MNI stereotactic space, and (5) studies on DD in the first language. The exclusion criteria were (1) studies with only ROI analysis, (2) resting-state studies, (3) studies that only included readers with DD or did not report group differences, (4) studies with direct group comparisons only between readers with DD and reading level control readers, (5) studies on children at risk for dyslexia, and (6)

studies focused on non-linguistic tasks (*Evans et al., 2014b*; *Margolis et al., 2020*; *Menghini et al., 2006*; *Yang et al., 2013*). Finally, 119 experiments from 110 papers were included in this meta-analysis comprising 92 brain functional experiments (from 87 papers) and 27 brain structural experiments (from 23 papers) (see *Table 9*; *Table 10* and *Figure 4* for detail). From the original publications, we extracted peak coordinates, where there is a significant difference between controls and individuals with DD either in brain activation or regional GMV. We also extracted effect sizes and other information from the publications. In order to explore the language effect, we subdivided these studies into two groups according to the native language of the participants, namely, an alphabetic language group in which writing symbols represent phonemes, and a morpho-syllabic language group in which each writing symbol represents a morpheme with a syllable. This procedure resulted in 79 functional and 21 structural experiments for the alphabetic language group and 12 functional and six structural experiments for the morpho-syllabic language group.

## Voxel-wise meta-analysis

After data acquisition, we conducted a voxel-wise meta-analysis using the anisotropic effect-size version of Signed Differential Mapping software (AES-SDM version 5.14, see http://www.sdmproject.com) separately for functional studies and structural studies in alphabetic languages and morpho-syllabic languages. Unlike other coordinate-based meta-analysis methods such as Activation likelihood estimation (ALE) or Multilevel peak Kernel density analysis (MKDA), AES-SDM combined the peak coordinates with the statistical parameter maps to increase the sensitivity of the analysis (*Radua et al., 2012a*). Data were first preprocessed with the statistical parameter maps and the peak coordinates were convolved with a fully anisotropy un-normalized Gaussian kernel ($\alpha = 1$) (full width at half maximum = 20 mm) to recreate the effect size map and the corresponding variance map for each study (*Radua et al., 2012a*; *Radua et al., 2014*). Then, a random-effect model was set up to calculate the differences between the DD group and the control group. Five hundred permutations were performed to ensure the stability of the analysis. Finally, the results of the standard meta-analysis were thresholded at peak height of the mean effect size SDM-Z = 1, uncorrected p = 0.005 at the voxel level and 150 voxels at the cluster level, which is stricter than the threshold suggested by *Radua et al., 2012a* (peak height SDM-Z = 1, uncorrected p = 0.005 at the voxel level and 10 voxels at the cluster level) in order to avoid false-positive results and gain enough sensitivity.

In order to identify differences between the two language groups, we conducted a direct comparison between the alphabetic language group and the morpho-syllabic language group for functional studies and structural studies separately, using SDM linear model function. The threshold was set at peak height SDM-Z = 1, voxel level uncorrected p = 0.005 and 150 voxels at the cluster level.

To find out the common language difference between the structural and functional studies, we conducted a conjunction minimum analysis (*Friston et al., 1999*; *Nichols et al., 2005*) using the image calculation function of SPM12 (https://www.fil.ion.ucl.ac.uk) between the language differences in the structural studies and the language differences in the functional studies.

To test the stability of the meta-analysis results, we conducted a whole-brain jack-knife sensitivity analysis. The standard meta-analysis was repeated n times (n = 79 for functional experiments in alphabetic languages, n = 12 for functional experiments in morpho-syllabic languages, n = 21 for structural experiments in alphabetic languages, n = 6 for structural experiments in morpho-syllabic languages) but leaving out one experiments each time, to determine whether the results remained significant.

## Multimodal meta-analysis

Because we were interested in the convergence between functional deficits and structural deficits, a multimodal meta-analysis was conducted in alphabetic languages and morpho-syllabic languages separately, which provided an efficient way to combine two meta-analyses in different modalities. The union probabilities of the meta-analytical maps of functional studies and structural studies were estimated and then thresholded at the peak height p = 0.00025, with a voxel level uncorrected p = 0.0025 and 150 voxels at the cluster level, which was stricter than the one suggested by *Radua et al., 2012b*; *Radua et al., 2013* (peak height p = 0.00025, with a voxel level uncorrected p = 0.0025 and 10 voxels at cluster level).

To find out the common multimodal deficits in the two language groups, we conducted a conjunction minimum analysis (*Friston et al., 1999*; *Nichols et al., 2005*) using the image calculation function

**Table 9.** Functional studies included in the meta-analysis.

| Studies | N (TD) | N (DD) | Age in months | Language | Writing system | Subject type | Tasks | Voxel-wise | Cluster-wise |
|---|---|---|---|---|---|---|---|---|---|
| Bach et al., 2010 | 18 | 14 | 99.6 | German | Alphabetic | Children | Covert reading task | p < 0.005 | 24 voxels* |
| Beneventi et al., 2009 | 13 | 11 | 160.4 | Norwegian | Alphabetic | Children | Sequential verbal working memory task | p < 0.001 | 10 voxels |
| Beneventi et al., 2010a; | 13 | 11 | 160.4 | Norwegian | Alphabetic | Children | n-back task (Letter) | FDR p < 0.05 | 5 voxels |
| Beneventi et al., 2010b | 14 | 12 | 160.3 | Norwegian | Alphabetic | Children | n-back task (Picture) | FDR p < 0.05 | |
| Blau et al., 2009 | 13 | 13 | 301.8 | Dutch | Alphabetic | Adults | Letter–speech-sound integration task | p < 0.001 | 160 mm³* |
| Booth et al., 2007a | 13 | 13 | 126.0 | English | Alphabetic | Children | Word judgment task | p < 0.001 | 15 voxels |
| Boros et al., 2016 | 17 | 12 | 129.7 | French | Alphabetic | Children | String detection and passive reading task | p < 0.001 | FDR p < 0.05 |
| Brambati et al., 2006 | 11 | 13 | 368.5 | Italian | Alphabetic | Adults and adolescents | Word reading and pseudoword reading | p < 0.001 | 20 voxels |
| Brunswick et al., 1999 | 6 | 6 | 277.2 | English | Alphabetic | Adults | Explicit reading task | p < 0.001 | |
| Brunswick et al., 1999 | 6 | 6 | 294.0 | English | Alphabetic | Adults | Implicit reading task | p < 0.001 | |
| Cao et al., 2008 | 12 | 12 | 148.2 | English | Alphabetic | Children | Visual word rhyming task | p < 0.001 | 10 voxels |
| Cao et al., 2017 | 13 | 17 | 134.0 | Chinese | Morpho-syllabic | Children | Auditory rhyming task | p < 0.001 | FDR p < 0.05 |
| Cao et al., 2018 | 19 | 23 | 132.9 | Chinese | Morpho-syllabic | Children | Visual spelling task | p < 0.001 | FDR p < 0.05 |
| Cao et al., 2020 | 17 | 16 | 137.3 | Chinese | Morpho-syllabic | Children | Visual rhyming task | p < 0.001 | FDR p < 0.05 |
| Christodoulou et al., 2014 | 12 | 12 | 274.8 | English | Alphabetic | Adults | Sentence reading | p < 0.001 | FDR corrected |
| Chyl et al., 2019 | 24 | 24 | 105.4 | Polish | Alphabetic | Children | Visual word reading | p < 0.001 | 50 voxels* |
| Conway et al., 2008 | 11 | 11 | 420.0 | English | Alphabetic | Adults | Auditory working memory task | p < 0.005 | 150 mL |
| Cutting et al., 2013 | 19 | 20 | 147.02 | English | Alphabetic | Adolescents | Lexical decision task | p < 0.005 | 34 voxels |
| Danelli et al., 2017 | 23 | 20 | 250.55 | Italian | Alphabetic | Adults | Pseudoword reading, auditory letter-name rhyming task, visual motion stimulation task and motor sequence learning task | p < 0.001 | FWE p < 0.05 |
| Desroches et al., 2010 | 12 | 12 | 137.4 | English | Alphabetic | Children | Auditory rhyming task | p < 0.001 | 15 voxels |
| Dufor et al., 2007 | 16 | 14 | 344.6 | French | Alphabetic | Adult | Auditory phoneme categorization task | p < 0.01 | |
| Eden et al., 2004 | 19 | 19 | 512.4 | English | Alphabetic | Adults | Word repetition task and initial sound deletion task | p < 0.001 | 80 voxels |
| Farris et al., 2016 | 16 | 15 | 112.2 | English | Alphabetic | Children | Object rhyming task | p < 0.001 | 10 voxels** |
| Feng et al., 2017 | 20 | 14 | 123.1 | Chinese | Morpho-syllabic | Children | Character spelling task and character rhyming task | p < 0.001 | 12 voxels* |
| Francisco et al., 2018 | 20 | 21 | 303.7 | Dutch | Alphabetic | Adults | 1-back task | p < 0.001 | FWE p < 0.05 |
| Gaab et al., 2007 | 23 | 22 | 127.8 | English | Alphabetic | Children | Sound discrimination task | p < 0.01 | 20 voxels |
| Georgiewa et al., 1999 | 17 | 17 | 168.0 | German | Alphabetic | Children | Letter reading task, nonwords reading task, words reading task and phonological transformation task | p < 0.05 | p < 0.05 |
| Grande et al., 2011 | 25 | 20 | 115.1 | German | Alphabetic | Children | Picture naming task and words reading task | p < 0.001 | t10 voxels |

*Table 9 continued on next page*

Table 9 continued

| Studies | N (TD) | N (DD) | Age in months | Language | Writing system | Subject type | Tasks | Threshold Voxel-wise | Cluster-wise |
|---|---|---|---|---|---|---|---|---|---|
| Grunling et al., 2004 | 21 | 17 | 162.8 | German | Alphabetic | Adolescents | Slash patterns matching task, letters matching task, words matching task, pseudoword matching task and pseudoword rhyming task | p < 0.01 | 10 voxels |
| Hancock et al., 2016 | 11 | 16 | 125.0 | English | Alphabetic | Children | Word rhyming task | p < 0.01 | 50 voxels |
| Heim et al., 2010 | 20 | 16 | 113.9 | German | Alphabetic | Children | First sound detection task, motion detection task, Posner attention task, auditory discrimination task | p < 0.001 | |
| Heim et al., 2013 | 15 | 11 | 435.4 | German | Alphabetic | Adults | Overt word reading | p < 0.05 | p < 0.05 |
| Heim et al., 2015 | 10 | 33 | 118.9 | German | Alphabetic | Children | Overt word reading | FWE p < 0.05 | 100 voxels |
| Hernandez et al., 2013 | 16 | 15 | 252.7 | French | Alphabetic | Adults | Word rhyming task and font matching task | p < 0.001* | |
| Higuchi et al., 2020 | 14 | 11 | 172.7 | Japanese | Morpho-syllabic | Adolescents | Character/picture passive viewing task | p < 0.005 | 20 voxels |
| Hoeft et al., 2006 | 10 | 10 | 133.9 | English | Alphabetic | Children | Visual word rhyming task | p < 0.001 | 10 voxels |
| Hoeft et al., 2007 | 19 | 19 | 172.8 | English | Alphabetic | Adolescents | Visual word rhyming task | p < 0.001 | 10 voxels |
| Horowitz-Kraus et al., 2016 | 9 | 10 | 120.5 | English | Alphabetic | Children | Narrative comprehension task | p < 0.001 | FWE corrected |
| Hu et al., 2010 | 8 | 8 | 171.6 | Chinese | Morpho-syllabic | Children | Sematic match task, word/ picture naming task | p < 0.001* | |
| Hu et al., 2010 | 10 | 11 | 164.5 | English | Alphabetic | Children | Sematic match task, word/ picture naming task | p < 0.001* | |
| Ingvar et al., 2002 | 9 | 9 | 287.0 | Swedish | Alphabetic | Adults | Word reading task and nonword reading task | p < 0.001 | |
| Jaffe-Dax et al., 2018 | 19 | 20 | 302.2 | Hebrew | Alphabetic | Adults | Tone frequency discrimination task | | p < 0.05 corrected |
| Kast et al., 2011 | 13 | 12 | 314.5 | German | Alphabetic | Adults | Lexical decision task | p < 0.001 | 30 voxels |
| Kovelman et al., 2012 | 12 | 12 | 108.4 | English | Alphabetic | Children | Auditory words rhyming task and auditory words matching task | p < 0.001 | 25 voxels* |
| Kronbichler et al., 2006 | 15 | 13 | 187.9 | German | Alphabetic | Adolescents | Sentence verification task | FDR p < 0.05 | 4 voxels |
| Kronschnabel et al., 2013 | 22 | 13 | 191.7 | German | Alphabetic | Adolescents | Rapid serial visual stimulation detect task | p < 0.005 | 160 voxels* |
| Kronschnabel et al., 2014 | 22 | 13 | 190.9 | German | Alphabetic | Adolescents | Target detection task | p < 0.005 | 160 voxels* |
| Landi et al., 2010 | 13 | 13 | 157.8 | English | Alphabetic | Adolescents | Rhyming task and semantic categorization task | FDR p < 0.01 | 20 voxels |
| Langer et al., 2015 | 15 | 15 | 119.4 | English | Alphabetic | Children | Sentence reading task | p < 0.005 | 50 voxels |
| Liu et al., 2012 | 11 | 11 | 142.8 | Chinese | Morpho-syllabic | Children | Word rhyming task and semantic judgment task | p < 0.0002 | 18 voxels* |
| Liu et al., 2013a | 14 | 14 | 141.8 | Chinese | Morpho-syllabic | Children | Lexical match task and character rhyming task | p < 0.001 | 10 voxels |
| Lobier et al., 2014 | 12 | 12 | 129.6 | French | Alphabetic | Adults | visual categorization of character task | p < 0.001 | 20 voxels |
| MacSweeney et al., 2009 | 7 | 7 | 343.5 | English | Alphabetic | Adults | Picture rhyming task | p < 0.01 | 20 voxels |
| Maurer et al., 2011 | 16 | 11 | 136.6 | German | Alphabetic | Children | Word matching task, pseudoword matching task, picture matching task | p < 0.01 | 30 voxels* |
| McCrory et al., 2000 | 6 | 8 | 275.0 | English | Alphabetic | Adults | Words and pseudowords production | p < 0.001, | |

# Table 9 continued

| Studies | N (TD) | N (DD) | Age in months | Language | Writing system | Subject type | Tasks | Threshold Voxel-wise | Cluster-wise |
|---|---|---|---|---|---|---|---|---|---|
| McCrory et al., 2005 | 10 | 8 | 242.0 | English | Alphabetic | Adults | Words reading and pictures naming | p < 0.05 corrected | |
| Meyler et al., 2008 | 12 | 23 | 129.6 | English | Alphabetic | Children | Sentence comprehension | p < 0.002 | 10 voxels |
| Monzalvo et al., 2012 | 23 | 23 | 130.0 | French | Alphabetic | Children | Passive picture/word viewing task and passive sentence listening task | p < 0.001 | p < 0.05 corrected |
| Olulade et al., 2012 | 9 | 6 | 247.7 | English | Alphabetic | Adults | Word rhyme task 3-D spatial rotations | p < 0.005 | 10 voxels |
| Olulade et al., 2015 | 12 | 16 | 120.5 | English | Alphabetic | Children | Implicit word reading | p < 0.001 | 20 voxels |
| Paulesu et al., 2001 | 36 | 36 | 286.4 | English, French, Italian | Alphabetic | Adults | Word and non-word reading task | p< 0.001 | corrected |
| Paulesu et al., 1996 | 5 | 5 | 314.5 | English | Alphabetic | Adults | Letter rhyming and letter memory | p < 0.001 | |
| Pecini et al., 2011 | 13 | 13 | 276.0 | Italian | Alphabetic | Adults and adolescents | Rhyme-generation task | p < 0.05 corrected | 100 mm³ |
| Pekkola et al., 2006 | 10 | 10 | 330.6 | Finnish | Alphabetic | Adults | Audio-visual speech perception | z > 1.8 | p < 0.05 corrected |
| Perrachione et al., 2016 | 19 | 19 | 279.6 | English | Alphabetic | Adults | Speech perception | p < 0.001 | p < 0.001 FDR |
| Perrachione et al., 2016 | 24 | 23 | 267.0 | English | Alphabetic | Adults | Spoken words listening, Written words, objects, and faces viewing | p < 0.001 | p < 0.001 FDR |
| Perrachione et al., 2016 | 25 | 26 | 95.4 | English | Alphabetic | Children | Spoken words listening, Written words, objects, and faces viewing | p < 0.001 | p < 0.05 FDR |
| Peyrin et al., 2011 | 12 | 12 | 120.0 | French | Alphabetic | Children | Categorical matching task | p < 0.001 | 15 voxels |
| Prasad et al., 2020 | 15 | 16 | 144.0 | Hindi | Syllabic | Children and adolescents | Auditory rhyming task, picture-naming task and semantic tasks | p < 0.001 | |
| Reilhac et al., 2013 | 12 | 12 | 306.6 | French | Alphabetic | Adults | Perceptual matching task | p < 0.001 | 15 voxels |
| Richlan et al., 2010 | 18 | 15 | 215.8 | German | Alphabetic | Adults and adolescents | phonological decision task | p < 0.005 | 20 voxels |
| Rimrodt et al., 2009 | 15 | 14 | 141.0 | English | Alphabetic | Children | Sentence comprehension task | p < 0.001 | 78 voxels* |
| Ruff et al., 2002 | 11 | 6 | 348.7 | French | Alphabetic | Aadults | Passive listening task | p < 0.01 | 38 voxels |
| Rumsey et al., 1997 | 14 | 17 | 313.2 | English | Alphabetic | Adults | Pronunciation task and lexical decision task | p < 0.01 | |
| Schulz et al., 2008 | 21 | 12 | 137.7 | German | Alphabetic | Children | Sentence reading task | p < 0.001 | |
| Schulz et al., 2009 | 15 | 15 | 138.0 | German | Alphabetic | Children | Sentence reading task | p < 0.001 | |
| Siok et al., 2004 | 8 | 8 | 132.0 | Chinese | Morpho-syllabic | Children | Homophone judgement task and lexical decision task | p < 0.001 | 20 voxels |
| Siok et al., 2008 | 12 | 12 | 131.5 | Chinese | Morpho-syllabic | Children | Character rhyming task | p < 0.005 | 10 voxels |
| Siok et al., 2009 | 12 | 12 | 131.5 | Chinese | Morpho-syllabic | Children | Font size judgment task | p < 0.05 FDR corrected | 10 voxels |
| Steinbrink et al., 2012 | 12 | 14 | 223.7 | German | Alphabetic | Adults and adolescents | Syllable discrimination | p < 0.05 FWE corrected | |
| Temple et al., 2000 | 10 | 8 | 362.7 | English | Alphabetic | Adults | Pitch discrimination task | p < 0.001 | |
| Temple et al., 2001 | 15 | 24 | 127.5 | English | Alphabetic | Children | Letter rhyming task, letter matching task | p < 0.001 | 20 voxels |
| van der Mark et al., 2009 | 24 | 18 | 136.1 | German | Alphabetic | Children | Phonological lexical decision task | p < 0.001 | 10 voxels |

Table 9 continued on next page

*Table 9 continued*

| Studies | N (TD) | N (DD) | Age in months | Language | Writing system | Subject type | Tasks | Threshold Voxel-wise | Cluster-wise |
|---|---|---|---|---|---|---|---|---|---|
| **van Ermingen-Marbach et al., 2013a** | 13 | 17 | 117.2 | German | Alphabetic | Children | Phoneme detection task | p < 0.001 | 10 voxels |
| **van Ermingen-Marbach et al., 2013a** | 13 | 14 | 116.4 | German | Alphabetic | Children | Phoneme detection task | p < 0.001 | 10 voxels |
| **van Ermingen-Marbach et al., 2013b** | 10 | 32 | 117.0 | German | Alphabetic | Children | Initial phoneme deletion task | p < 0.01 | 30 voxels |
| **Vasic et al., 2008** | 13 | 12 | 219.6 | German | Alphabetic | Adults | Verbal working memory task | p < 0.005 | p < 0.05 |
| **Waldie et al., 2013** | 16 | 12 | 365.1 | English | Alphabetic | Adults | Go/no-go lexical decision task | p < 0.001 | |
| **Weiss et al., 2016** | 22 | 21 | 325.1 | Hebrew | Alphabetic | Adults | Word reading task | p < 0.001 | 50 voxels |
| **Wimmer et al., 2010** | 19 | 20 | 247.6 | German | Alphabetic | Adults and adolescents | Phonological lexical decision task | p < 0.005 | 10 voxels |
| **Yang and Tan, 2020** | 16 | 16 | 123.5 | Chinese | Morpho-syllabic | Children | Homophone judgments task and component judgments task | p < 0.005 | 25 voxels* |
| **Zuk et al., 2018** | 13 | 11 | 114.0 | English | Alphabetic | Children | First sound matching task | p < 0.005 | 50 voxels |

*equivalent to p < 0.05 corrected; ** equivalent to p < 0.01 corrected.

**Table 10.** Structural studies included in the meta-analysis.

| Study | N(TD) | N(DD) | Age in months | Language | Writing system | Subject type | Threshold Voxel-wise | Cluster-wise |
|---|---|---|---|---|---|---|---|---|
| Adrian-Ventura et al., 2020 | 12 | 13 | 146.2 | Spanish | Alphabetic | Children | p < 0.05 FWE corrected | 750 voxels |
| Brambati et al., 2004 | 11 | 10 | 352.8 | Italian | Alphabetic | Adults and adolescents | p < 0.05 corrected for small brain volume | |
| Brown et al., 2001 | 14 | 16 | 288.0 | English | Alphabetic | Adults | p < 0.05 | p < 0.05 |
| Eckert et al., 2005 | 13 | 13 | 136.5 | English | Alphabetic | Children | p < 0.001 | |
| Evans et al., 2014a | 14 | 14 | 505.8 | English | Alphabetic | Adults | Pp< 0.001 | p < 0.05 |
| Evans et al., 2014a | 13 | 13 | 371.4 | English | Alphabetic | Adults | p < 0.001 | p< 0.05 |
| Evans et al., 2014a | 15 | 15 | 107.4 | English | Alphabetic | Children | p < 0.001 | p < 0.05 |
| Evans et al., 2014a | 17 | 17 | 115.2 | English | Alphabetic | Children | p < 0.001 | p < 0.05 |
| Hoeft et al., 2007 | 19 | 19 | 172.8 | English | Alphabetic | Children and adolescents | p < 0.01 | p < 0.01 |
| Jager-Rickels et al., 2018 | 32 | 17 | 114.2 | English | Alphabetic | Children | p < 0.001 | |
| Jednorog et al., 2014 | 35 | 46 | 123.5 | Polish | Alphabetic | Children | p < 0.001 | p < 0.05 |
| Jednorog et al., 2015 | 106 | 130 | 123.9 | French, German, Polish | Alphabetic | Children | p < 0.001 | 150 voxels* |
| Krafnick et al., 2014 | 15 | 15 | 118.2 | English | Alphabetic | Children | Pp< 0.01 | p < 0.01 FWE corrected |
| Kronbichler et al., 2008 | 15 | 13 | 187.9 | German | Alphabetic | Adolescents | p < 0.005 | |
| Liu et al., 2013b | 18 | 18 | 141.4 | Chinese | Morpho-syllabic | Cchildren | p < 0.001 | 196 voxels |
| Menghini et al., 2008 | 10 | 10 | 489.0 | Italian | Alphabetic | Adult | p < 0.005 | p < 0.05 corrected |
| Moreau et al., 2019 | 12 | 12 | 352.2 | English | Alphabetic | Adult | p < 0.05 FWE corrected | |
| Pernet et al., 2009 | 39 | 38 | 336.0 | French | Alphabetic | Adult | p< 0.05 FDR corrected | |

*Table 10 continued on next page*

*Table 10 continued*

| Study | N(TD) | N(DD) | Age in months | Language | Writing system | Subject type | Threshold Voxel-wise | Cluster-wise |
|---|---|---|---|---|---|---|---|---|
| *Silani et al., 2005* | 32 | 32 | 304.5 | Italian, French, English | Alphabetic | Adults | | |
| *Siok et al., 2008* | 16 | 16 | 132.0 | Chinese | Morpho-syllabic | Children | p < 0.05 FWE corrected | 50 voxels |
| *Steinbrink et al., 2008* | 8 | 8 | 262.8 | German | Alphabetic | Adults | p < 0.05 FDR corrected | |
| *Tamboer et al., 2015* | 57 | 37 | 245.3 | Dutch | Alphabetic | Adults | p < 0.05 | 200 voxels |
| *Vinckenbosch et al., 2005* | 10 | 13 | 282.0 | French | Alphabetic | Adults | p < 0.01 | p < 0.05 |
| *Wang et al., 2019* | 17 | 27 | 134.0 | Chinese | Morpho-syllabic | Children | p < 0.001 | p < 0.05 FWE corrected |
| *Xia et al., 2016* | 12 | 12 | 132.0 | Chinese | Morpho-syllabic | Children | p < 0.001 | |
| *Xia et al., 2016* | 12 | 12 | 169.2 | Chinese | Morpho-syllabic | Children | p < 0.001 | |
| *Yang et al., 2016* | 14 | 9 | 149.0 | Chinese | Morpho-syllabic | Children | p < 0.001 | 111 voxels* |

* equivalent to 0.05 corrected; ** equivalent to 0.01 corrected.

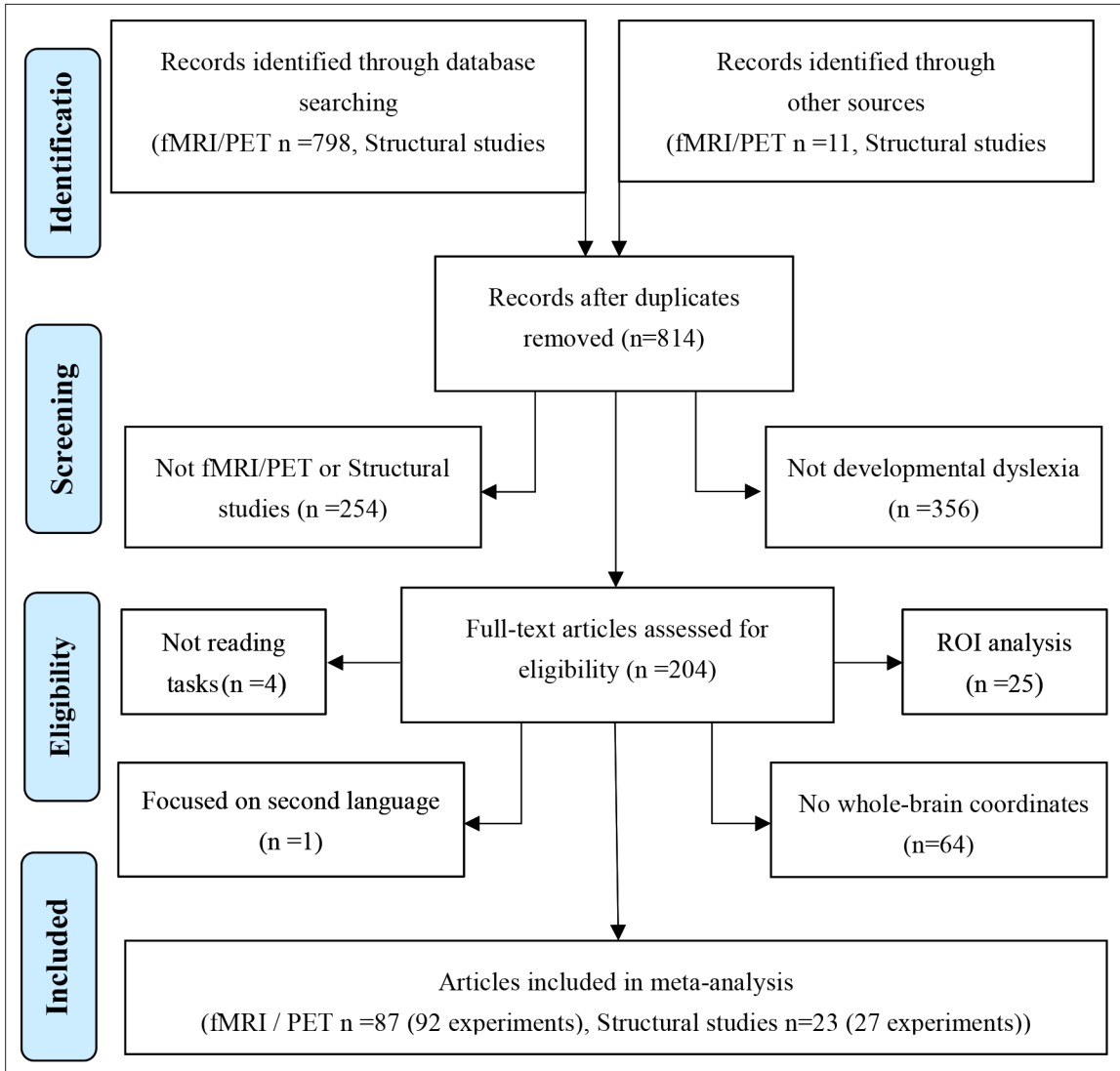

**Figure 4.** PRISMA flowchart of the selection process for included articles.

of SPM12 (https://www.fil.ion.ucl.ac.uk) between the multimodal deficits in the alphabetic group and the morpho-syllabic group.

## Confirmation study

Because there were studies on both adults and children in the alphabetic group, whereas most of the morpho-syllabic studies were on children, the language difference may be due to the unmatched age range in the two groups of studies. In order to eliminate the influence of the confound, we conducted a confirmation analysis with only studies on children in the alphabetic group (n = 36 for the functional studies and n = 8 for the structure studies). Then, we compared the alphabetic and morpho-syllabic groups for functional studies and structural studies separately.

We conducted another confirmation analysis with 10 English studies (*Booth et al., 2007a*; *Cao et al., 2008*; *Farris et al., 2016*; *Hancock et al., 2016*; *Hu et al., 2010*; *Langer et al., 2015*; *Meyler et al., 2008*; *Olulade et al., 2015*; *Rimrodt et al., 2009*; *Temple et al., 2001*) and 10 Chinese studies (*Cao et al., 2018*; *Cao et al., 2020*; *Feng et al., 2017*; *Hu et al., 2010*; *Liu et al., 2012*; *Liu et al., 2013a*; *Siok et al., 2004*; *Siok et al., 2008*; *Siok et al., 2009*; *Yang and Tan, 2020*), because different languages were included in the alphabetic language group and orthographic depth makes a difference as found in previous research (*Martin et al., 2016*). Therefore, we only included English studies in the alphabetic group in this confirmation analysis. The two subgroups were also matched

on participants' age (mean age = 10.95 years for English studies, mean age = 11.40 years for Chinese studies), number of studies and task (visual word tasks). Then, we conducted a direct comparison between the Chinese studies and English studies. The threshold was set at peak height SDM-Z = 1 and voxel level uncorrected p = 0.005 with 150 voxels at the cluster level.

## Acknowledgements

This work was supported by the "Fundamental Research Funds for the Central Universities" awarded to Dr. Fan Cao, "Guangdong Planning Office of Philosophy and Social Science" (GD19CXL05) awarded to Dr. Fan Cao, "Science and Technology Program of Guangzhou, China, Key Area Research and Development Program (202007030011)", and by "The national social science fund of China" (21BYY204) awarded to Dr. Fan Cao.

## Additional information

### Funding

| Funder | Grant reference number | Author |
|---|---|---|
| Fundamental Research Funds for the Central Universities | | Fan Cao |
| Guangdong Planning Office of Philosophy and Social Science | GD19CXL05 | Fan Cao |
| Science and Technology Program of Guangzhou, China, Key Area Research and Development Program | 202007030011 | Fan Cao |
| The National Social Science Fund of China | 21BYY204 | Fan Cao |

The funders had no role in study design, data collection and interpretation, or the decision to submit the work for publication.

### Author contributions

Xiaohui Yan, Data curation, Methodology, Visualization, Writing – original draft; Ke Jiang, Hui Li, Ziyi Wang, Data acquisition, Data acquisition, Data acquisition, Validation; Kyle Perkins, Writing – review and editing; Fan Cao, Conceptualization, Funding acquisition, Supervision, Writing – review and editing, Writing – original draft

### Author ORCIDs

Xiaohui Yan (iD) http://orcid.org/0000-0002-2801-3335
Fan Cao (iD) http://orcid.org/0000-0002-3786-1600

### Decision letter and Author response

Decision letter https://doi.org/10.7554/eLife.69523.sa1
Author response https://doi.org/10.7554/eLife.69523.sa2

## Additional files

### Supplementary files

• Supplementary file 1. Additional results for the meta-analysis. (a) Functional deficits in children with DD in alphabetic languages (b) Structural deficits in children with DD in alphabetic languages (c) Direct comparisons between functional studies in alphabetic languages and morpho-syllabic languages for children with DD (d) Direct comparisons between structural studies in alphabetic languages and morpho-syllabic languages for children with DD (e) Language differences in functional studies between the two well-matched groups in the confirmation analysis (f) Abnormal brain

functions found in the current study that were reported in each functional study of the alphabetic language group (g) Abnormal brain structures found in the current study that were reported in each structural study of the alphabetic group (h) Abnormal brain functions found in the current study that were reported in each functional study of the morpho-syllabic group (i) Abnormal brain structures found in the current study that were reported in each structural study of the morpho-syllabic group.

- Supplementary file 2. Key words used in literature retrieval.
- Transparent reporting form

### Data availability
All data generated or analysed during this study are included in the manuscript and supporting files. Meta-analysis data is deposited to Dryad.

The following dataset was generated:

| Author(s) | Year | Dataset title | Dataset URL | Database and Identifier |
|-----------|------|---------------|-------------|-------------------------|
| Yan X | 2021 | meta-analysis data | https://doi.org/10.5061/dryad.0p2ngf222 | Dryad Digital Repository, 10.5061/dryad.0p2ngf222 |

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
