## [Decision Letter]

**Acceptance summary:**

This is a rigorously conducted meta-analytic study investigating the functional and structural abnormalities associated with developmental dyslexia across languages. Convergent and divergent functional and structural changes as well as language-universal and language-specific brain alternations related to dyslexia are reported. The identification of the universal and language-specific neural manifestations of dyslexia is valuable, and it can help diagnose individuals with dyslexia in the given language with higher accuracy and administer effective interventions.

**Decision letter after peer review:**

Thank you for submitting your article "Convergent and divergent structural and functional brain abnormalities associated with developmental dyslexia: a cross-linguistic meta-analysis of neuroimaging studies" for consideration by *eLife*. Your article has been reviewed by 2 peer reviewers, and the evaluation has been overseen by a Reviewing Editor and Chris Baker as the Senior Editor. The following individual involved in review of your submission has agreed to reveal their identity: Li Hai Tan (Reviewer #1).

Essential revisions:

Hereby are summarized the main essential revisions for the papers from the two reviewers. More detailed revisions and changes are included below in each review:

1) The meta-analysis pooling all studies/experiments together seem unnecessary and may even harm this study. Thus, Reviewer 2 suggests to carefully think it over.

2) As mentioned also in Reviewer 2 comments as well, they highly recommend considering the age effect and give a detailed discussion.

3) The LIFG is a big region. Reviewer 1 and myself consider It would be helpful to more precisely present the findings related to this region, for example, by using the dorsal IFG and the ventral IFG. This will help readers more quickly understand the universality and the specificity of cortical regions across languages.

4) There are several methodological details that need to be included in the manuscript as outlined in Reviewer 2 comments.

5) Some additional points need to be discussed in the manuscript or rephrased/removed as detailed in both reviewers' comments. In that sense, the authors should be careful since some statements are out of the scope of this study and can be misleading (see Reviewer 2 comments).

6) Reviewer 2 gives some suggestions to restructure the information included in the introduction and discussion in order to convey a clearer message since there is too much information included in the introduction.

*Reviewer #1 (Recommendations for the authors):*

1. For the meta-analysis results, it would be helpful to show the brain maps of functional and structural impairment associated with DD in the main text instead of in supplement. The same for brain regions that had normal structure but altered function. All these brain maps can be added in Figure 2.

2. The results showing divergent structural and functional alterations are very interesting. In addition to the reasons the authors mentioned in the paper, another possibility is that the brain changes slowly but the reading performance changes rapidly – the quick development of reading skills may lead to a dissociation of structure and function. A recent study by Siok et al., (Cerebral Cortex, 2020) showed this dissociation.

3. The model by Friederici (2012) is important; it seems to involve the ventral portion of the IFG, not the dorsal portion. The author should be more precise in the paper.

*Reviewer #2 (Recommendations for the authors):*

Title

1. It is unclear what the focus is-convergence and divergence between functional and structural abnormalities or between different languages, or both.

Introduction

2. General:

(a) Given that the main aim of the meta-analysis is to synthesize the previous finding, it is better to summarize the previous studies in the introduction and present detailed explanations in the discussion. In the current version, too many details are presented in the introduction. For example, discussion about the language-specific in neural impairments in dyslexia (p. 5-6, lines 96-128) and the difference between Chinese and alphabetic languages (p. 7-8, lines 129-153).

(b) There are meta-analysis studies that focus on different aspects, some of which are relevant to the authors' question in this study. Thus, these works should be mentioned, and key findings should be summarized.

3. p. 3-4, lines 43-59: The authors use phonological deficits as an example to explain the manifestation of the "common" impairments can be influenced by linguistic features. While it is a good example, it is also important to notice that nowadays a large body of evidence support a multi-deficit hypothesis (where the phonological deficit is one of the most common one) and the "neural abnormalities" could be associated with different cognitive deficits.

4. p. 4, line 60: Richlan updated the model in his 2014 Fronts Hum Neurosci paper.

5. p. 4, lines 75-77: In Richlan 2014, Fronts Hum Neurosci, the left IFG and preCG have been highlighted with distinct functions. Please also see Hancock et al., 2017, Neur Biobehav Rev.

6. p. 5, lines 86-87: It is not the number of studies finding language-universal deficits associated with dyslexia is few. In fact, the research investigating this topic is lacking, to which, in my opinion, meta-analysis can contribute.

7. p. 8, line 154: Before introducing structure MRI studies, it may be better to declare the significance of conducting structural imaging research.

8. p. 8-9, lines 163-170: First, it may be better to discuss studies in Chinese and alphabetic languages separately. Second, it is true that these regions do not belong to the "three most reported brain regions involved in reading processing." However, they may also be part of the network recruited for specific reading processes.

9. p. 9, lines 174-177: It is possible that a brain measure not associated with reading experiences can also manifest to be language-specific. For example, it could be correlated with cognitive processing required by reading in a specific language.

Methods

10. To deal with the problem caused by mixing alphabetic and morpho-syllabic languages, besides conducting analysis separately for each category of the writing system, another way is to treat writing system type as modality, i.e., conduct "multi-modal like" analysis.

11. As for the age effect, focusing on children in the primary analysis could be an option.

12. Explicitly and clearly list the combinations of the keywords. E.g., "dyslexia" and "fMRI"; "dyslexia" and "PET"; etc.

13. p. 11, line 212: Can the authors list studies adopting a "volumetric FreeSurfer pipeline"? To my knowledge, such a pipeline is commonly used in investigating subcortical structures.

14. The threshold for dealing with whole-brain multiple comparisons error. As mentioned by Muller et al., 2018 Neur Biobehav Rev, a formal threshold (e.g., p-voxel < 0.001, FWE corrected p-cluster < 0.05) is recommended even for ES-SDM/AES-SDM.

Results

15. p. 19, lines 363-367: Please present more details about the "conjunction" analysis?

16. p. 19, line 366: What does the peak mean, the center of gravity?

17. I suggest adding information about whether the specific deficit was observed in each included study.

Discussion

18. p. 22, line 417: It is convergence/divergence between functional and structural differences between dyslexia and control, rather than the "relationship." There is no evidence on whether these deficits are related to each other.

19. p. 23, lines 440-442: The authors mentioned, "The left STG and the dorsal IFG are two core regions in the language model proposed by Friederici (2012) and by Hickok and Poeppel (2007)". How about the reading models?

20. p. 28-29, lines 553-581: The authors interpret the increased GMV and hyper-activation in dyslexia as compensation. However, there might be other possibilities (e.g., it could be another type of deficits in dyslexia). In addition, the authors may discuss what analysis (e.g., brain-behavior correlation, longitudinal design, etc.) can be done further to examine these possibilities.

21. p. 32, lines 646-648: "… the finding of neurological alterations in the cerebellum supports thee cerebellar deficit hypothesis." Can the authors discuss what the cerebellar deficit hypothesis claims and how the current findings support them?

22. p. 33, lines 663: "…, which is consistent with our finding" This is what a meta-analysis does. A better approach could be listing studies that observed and did not observe such a pattern.

Limitations

23. Each technique has its strength and weakness, while they can all contribute to depicting the whole picture. The main aim of an fMRI/sMRI/PET meta-analysis is to synthesize existing evidence. Therefore, in my opinion, "the neurophysiology of changes in brain structure and function is unknown" is not a limitation of this study. In synthesizing the previous findings, the authors recreated the statistical maps, which can be more sensitive than other coordinate-based meta-analytic approaches. Even though, analysis with the original statistical maps will be beneficial. These maps can be acquired by requiring the authors of the original studies or downloading from neural data sharing platforms such as Neurovault (https://neurovault.org/). Second, as mentioned before, several factors (age, orthographic depth, fMRI task paradigm, etc.) are not considered in the current analysis. Third, diffusion MRI and resting-state fMRI are important parts of neuroimaging study on dyslexia. Given that these studies have their characteristics in terms of data analysis and result reports (e.g., fiber tracking), as well as a relatively limited number, it is hard to run mete-analysis. Nevertheless, it is informative to include these studies when discussing the current findings and future directions.

Table

24. Table 1: Multiple correction methods need to be included.

Figures

25. General:

(a) The appearance of each cluster looks more extensive on the standard brain given the size of the cluster. What the voxel size and the total number of the entire brain are?

(b) Please add legends to each figure, which can help readers understand the information conveyed well.

(c) Please add color bars if applicable.

26. Figure 1: Please add indicators (e.g., identification, screening, eligibility, included) for different portions.

27. Since there were also findings in the subcortical areas (e.g., the bilateral caudate nucleus in Figure S1), it will be helpful to include a slice view in relevant figures.

28. Figures S3-S6 are not mentioned in the main text.

---

## [Author Response]

Essential revisions:Hereby are summarized the main essential revisions for the papers from the two reviewers. More detailed revisions and changes are included below in each review:1) The meta-analysis pooling all studies/experiments together seem unnecessary and may even harm this study. Thus, Reviewer 2 suggests to carefully think it over.

We agree with the reviewers. We deleted the meta-analysis on all studies/experiments. Now we only have analysis on each language group, and the direct comparisons as well as conjunctions between them.

2) As mentioned also in Reviewer 2 comments as well, they highly recommend considering the age effect and give a detailed discussion.

Following the reviewer’s suggestion, we conducted a confirmation analysis by comparing children in the alphabetic group and children in the morpho-syllabic group. Results are consistent with the original findings. We added discussion on page 21, 22.

3) The LIFG is a big region. Reviewer 1 and myself consider It would be helpful to more precisely present the findings related to this region, for example, by using the dorsal IFG and the ventral IFG. This will help readers more quickly understand the universality and the specificity of cortical regions across languages.

We differentiate IFG into orbital, triangular, and opercular parts in the manuscript now. The IFG where we found greater deficit in morpho-syllabic languages than alphabetic languages is more dorsal than and overlaps with the IFG opercular part.

4) There are several methodological details that need to be included in the manuscript as outlined in Reviewer 2 comments.

We added details in the method to address the reviewer’s comments.

5) Some additional points need to be discussed in the manuscript or rephrased/removed as detailed in both reviewers' comments. In that sense, the authors should be careful since some statements are out of the scope of this study and can be misleading (see Reviewer 2 comments).

We carefully addressed reviewer 2’s comments.

6) Reviewer 2 gives some suggestions to restructure the information included in the introduction and discussion in order to convey a clearer message since there is too much information included in the introduction.

We made the introduction more concise.

Reviewer #1 (Recommendations for the authors):1. For the meta-analysis results, it would be helpful to show the brain maps of functional and structural impairment associated with DD in the main text instead of in supplement. The same for brain regions that had normal structure but altered function. All these brain maps can be added in Figure 2.

We added these maps in Figure 1.

2. The results showing divergent structural and functional alterations are very interesting. In addition to the reasons the authors mentioned in the paper, another possibility is that the brain changes slowly but the reading performance changes rapidly – the quick development of reading skills may lead to a dissociation of structure and function. A recent study by Siok et al., (Cerebral Cortex, 2020) showed this dissociation.

We appreciate the reviewer’s thoughtful suggestion. We added this argument and reference on page 8-9, 26.

3. The model by Friederici (2012) is important; it seems to involve the ventral portion of the IFG, not the dorsal portion. The author should be more precise in the paper.

We reorganized this paragraph in the discussion now with a focus on STG, since that is where the convergence between structural and functional findings across languages is. Please see page 19, 20.

Reviewer #2 (Recommendations for the authors):Title1. It is unclear what the focus is-convergence and divergence between functional and structural abnormalities or between different languages, or both.

It is both. We focused on the convergence and divergence between functional and structural brain alterations associated with dyslexia and whether the convergence/divergence pattern is the same or different in different languages. We changed the title to make it clearer.

Introduction2. General:(a) Given that the main aim of the meta-analysis is to synthesize the previous finding, it is better to summarize the previous studies in the introduction and present detailed explanations in the discussion. In the current version, too many details are presented in the introduction. For example, discussion about the language-specific in neural impairments in dyslexia (p. 5-6, lines 96-128) and the difference between Chinese and alphabetic languages (p. 7-8, lines 129-153).

We revised the introduction to make it more concise on page 5, 6, 7, 8.

(b) There are meta-analysis studies that focus on different aspects, some of which are relevant to the authors' question in this study. Thus, these works should be mentioned, and key findings should be summarized.

We added the meta-analysis studies related to our study on page 6, line 102-111, page 7, line 129-132, line 139-144.

3. p. 3-4, lines 43-59: The authors use phonological deficits as an example to explain the manifestation of the "common" impairments can be influenced by linguistic features. While it is a good example, it is also important to notice that nowadays a large body of evidence support a multi-deficit hypothesis (where the phonological deficit is one of the most common one) and the "neural abnormalities" could be associated with different cognitive deficits.

We reworded this part. Now we refer to the multi-deficit hypothesis, and added that phonological deficit is one of the deficits on page 3, line 42-45.

4. p. 4, line 60: Richlan updated the model in his 2014 Fronts Hum Neurosci paper.

We updated the model in our manuscript on page 4-5.

5. p. 4, lines 75-77: In Richlan 2014, Fronts Hum Neurosci, the left IFG and preCG have been highlighted with distinct functions. Please also see Hancock et al., 2017, Neur Biobehav Rev.

We now added the distinction between IFG and precentral gyrus on page 4-5.

6. p. 5, lines 86-87: It is not the number of studies finding language-universal deficits associated with dyslexia is few. In fact, the research investigating this topic is lacking, to which, in my opinion, meta-analysis can contribute.

We revised the sentence on page 5, line 91-92, 97-99.

7. p. 8, line 154: Before introducing structure MRI studies, it may be better to declare the significance of conducting structural imaging research.

We added the declaration on page 7, line 134-139.

8. p. 8-9, lines 163-170: First, it may be better to discuss studies in Chinese and alphabetic languages separately. Second, it is true that these regions do not belong to the "three most reported brain regions involved in reading processing." However, they may also be part of the network recruited for specific reading processes.

We revised this part. See page 7-8, line 139-153.

9. p. 9, lines 174-177: It is possible that a brain measure not associated with reading experiences can also manifest to be language-specific. For example, it could be correlated with cognitive processing required by reading in a specific language.

We reworded that sentence on page 8, line 156-158.

Methods10. To deal with the problem caused by mixing alphabetic and morpho-syllabic languages, besides conducting analysis separately for each category of the writing system, another way is to treat writing system type as modality, i.e., conduct "multi-modal like" analysis.

We conducted analyses separately for each category and then examined their conjunction. The "multi-modal like" analysis is actually the same as conjunction analysis in unimodal studies.

11. As for the age effect, focusing on children in the primary analysis could be an option.

We conducted a confirmation analysis to only include studies on children in the alphabetic group and the results replicated the findings when all studies were included, suggesting that language differences should not be driven by different age ranges in the two language groups.

12. Explicitly and clearly list the combinations of the keywords. E.g., "dyslexia" and "fMRI"; "dyslexia" and "PET"; etc.

We reworded the criteria of combinations of key words on page 31-32, line 612-627. We listed all the combinations we used in the Supplementary file 2.

13. p. 11, line 212: Can the authors list studies adopting a "volumetric FreeSurfer pipeline"? To my knowledge, such a pipeline is commonly used in investigating subcortical structures.

We included "volumetric FreeSurfer pipeline" as one of selection criteria on page 32. However, we didn’t find studies on DD using this pipeline.

14. The threshold for dealing with whole-brain multiple comparisons error. As mentioned by Muller et al., 2018 Neur Biobehav Rev, a formal threshold (e.g., p-voxel < 0.001, FWE corrected p-cluster < 0.05) is recommended even for ES-SDM/AES-SDM.

We carefully read through the paper by Muller et al., (Muller et al., 2018), but the formal threshold (e.g., p-voxel < 0.001, FWE corrected p-cluster < 0.05) is recommended for ALE meta-analysis. For ES-SDM/AES-SDM, it says “For ES-SDM, a previous simulation showed that an uncorrected threshold of p=0.005 with a cluster extent of 10 voxels and SDM-Z>1 adequately controlled the probability of detecting an effect by chance, and it is thus recommended (Radua et al., 2012). However, this is again an informal control of the false positive rate and could be too conservative or too liberal in other datasets, it must be understood as an approximation to corrected results.” Therefore, it does not necessarily mean that a more conservative threshold should be used for ES-SDM. Moreover, several recent meta-analysis papers also used the threshold suggested by Radua, 2012 (Pico-Perez et al., 2020; Schurz et al., 2021).

Results15. p. 19, lines 363-367: Please present more details about the "conjunction" analysis?

We added the details about the "conjunction analysis" on page 14 as well as in the methods section on page 35, 36.

16. p. 19, line 366: What does the peak mean, the center of gravity?

As we conducted a geometric average of the thresholded 1-p maps, the ‘peak’ means the maximum value of the averages.

17. I suggest adding information about whether the specific deficit was observed in each included study.

We added the information in the Supplementary file 1f-Supplementary file 1i. we also mentioned this information in the main text on page 12.

Discussion18. p. 22, line 417: It is convergence/divergence between functional and structural differences between dyslexia and control, rather than the "relationship." There is no evidence on whether these deficits are related to each other.

We revised the sentence on page 18, line 323-325.

19. p. 23, lines 440-442: The authors mentioned, "The left STG and the dorsal IFG are two core regions in the language model proposed by Friederici (2012) and by Hickok and Poeppel (2007)". How about the reading models?

We added that they were also key regions in the reading network on page 19, line 348-350.

20. p. 28-29, lines 553-581: The authors interpret the increased GMV and hyper-activation in dyslexia as compensation. However, there might be other possibilities (e.g., it could be another type of deficits in dyslexia). In addition, the authors may discuss what analysis (e.g., brain-behavior correlation, longitudinal design, etc.) can be done further to examine these possibilities.

We revised this section on page 25 to include the points raised by the reviewer.

21. p. 32, lines 646-648: "… the finding of neurological alterations in the cerebellum supports thee cerebellar deficit hypothesis." Can the authors discuss what the cerebellar deficit hypothesis claims and how the current findings support them?

We reworded this sentence on page 29.

22. p. 33, lines 663: "…, which is consistent with our finding" This is what a meta-analysis does. A better approach could be listing studies that observed and did not observe such a pattern.

We revised this part on page 30.

Limitations23. Each technique has its strength and weakness, while they can all contribute to depicting the whole picture. The main aim of an fMRI/sMRI/PET meta-analysis is to synthesize existing evidence. Therefore, in my opinion, "the neurophysiology of changes in brain structure and function is unknown" is not a limitation of this study. In synthesizing the previous findings, the authors recreated the statistical maps, which can be more sensitive than other coordinate-based meta-analytic approaches. Even though, analysis with the original statistical maps will be beneficial. These maps can be acquired by requiring the authors of the original studies or downloading from neural data sharing platforms such as Neurovault (https://neurovault.org/). Second, as mentioned before, several factors (age, orthographic depth, fMRI task paradigm, etc.) are not considered in the current analysis. Third, diffusion MRI and resting-state fMRI are important parts of neuroimaging study on dyslexia. Given that these studies have their characteristics in terms of data analysis and result reports (e.g., fiber tracking), as well as a relatively limited number, it is hard to run mete-analysis. Nevertheless, it is informative to include these studies when discussing the current findings and future directions.

We deleted “the neurophysiology of changes in brain structure and function is unknown” and added the lack of original statistical maps as a limitation on page 31. Second, we conducted confirmation analysis, and now these factors (age, task, orthographic depth) are taken into account when making conclusions. Third, we added findings from DTI and resting state studies to be more informative on page 20.

Table24. Table 1: Multiple correction methods need to be included.

We added the information in Table 1 and Table 2. See page 55-62.

Figures25. General:(a) The appearance of each cluster looks more extensive on the standard brain given the size of the cluster. What the voxel size and the total number of the entire brain are?

We used the default gray matter mask of AES-SDM software in this analysis, the voxel size was 2×2×2 mm^3^, and the total number of the brain were 193768 voxels. The extensive appearance of the clusters may be due to the software we used, as it presents all results onto the surface of the brain. To avoid this bias, we now use a slice view for all figures.

(b) Please add legends to each figure, which can help readers understand the information conveyed well.

We added legends.

(c) Please add color bars if applicable.

We added color bars.

26. Figure 1: Please add indicators (e.g., identification, screening, eligibility, included) for different portions.

We added indicators in Figure4 on page 33.

27. Since there were also findings in the subcortical areas (e.g., the bilateral caudate nucleus in Figure S1), it will be helpful to include a slice view in relevant figures.

We use a slice view for all figures now in the revised manuscript.

28. Figures S3-S6 are not mentioned in the main text.

We now mention all Tables and Figures in the revised manuscript.

References

Martin, A., Kronbichler, M., and Richlan, F. (2016). Dyslexic brain activation abnormalities in deep and shallow orthographies: A meta-analysis of 28 functional neuroimaging studies. Human Brain Mapping, 37(7), 2676-2699. doi:10.1002/hbm.23202

Muller, V. I., Cieslik, E. C., Laird, A. R., Fox, P. T., Radua, J., Mataix-Cols, D., Tench, C. R., Yarkoni, T., Nichols, T. E., Turkeltaub, P. E., Wager, T. D., and Eickhoff, S. B. (2018). Ten simple rules for neuroimaging meta-analysis. Neurosci Biobehav Rev, 84, 151-161. doi:10.1016/j.neubiorev.2017.11.012

Pico-Perez, M., Moreira, P. S., de Melo Ferreira, V., Radua, J., Mataix-Cols, D., Sousa, N., Soriano-Mas, C., and Morgado, P. (2020). Modality-specific overlaps in brain structure and function in obsessive-compulsive disorder: Multimodal meta-analysis of case-control MRI studies. Neurosci Biobehav Rev, 112, 83-94. doi:10.1016/j.neubiorev.2020.01.033

Schurz, M., Radua, J., Tholen, M. G., Maliske, L., Margulies, D. S., Mars, R. B., Sallet, J., and Kanske, P. (2021). Toward a hierarchical model of social cognition: A neuroimaging meta-analysis and integrative review of empathy and theory of mind. Psychological Bulletin, 147(3), 293-327. doi:10.1037/bul0000303